# Subspace Inference Enables Active Preference based Learning of Neural Network Reward Models

## Abstract

Reinforcement learning from human feedback (RLHF) has emerged as a powerful approach for aligning decision-making agents with human intentions, primarily through the use of reward models trained on human preferences. However, RLHF suffers from poor sample efficiency, as each feedback provides minimal information, making it necessary to collect large amounts of human feedback. Active learning addresses this by enabling agents to select informative queries, but effective uncertainty quantification required for active learning remains a challenge. While ensemble methods and dropout are popular for their simplicity, they are computationally expensive at scale and do not always provide good posterior approximation. Inspired by the recent advances in approximate Bayesian inference, we develop a method that leverages Bayesian filtering in neural network subspaces to efficiently maintain model posterior for active reward modeling. Our approach enables scalable sampling of neural network reward models to efficiently compute active learning acquisition functions. Experiments on the D4RL benchmark demonstrate that our approach achieves superior sample efficiency, scalability, and calibration compared to other Bayesian deep learning approaches, and leads to competitive offline reinforcement learning policy performance. This highlights the potential of scalable Bayesian methods for preference-based reward modeling in RLHF.

## 1 Introduction

In recent years, reinforcement learning from human feedback (RLHF) has become the dominant technique for aligning decision-making agents with human intentions (Christiano et al., 2017; Ouyang et al., 2022). The ease of providing preference feedback has been a crucial factor in their popularity as a feedback type for reward modeling, but since each feedback provides at most 1 bit of information, they are also known for their poor sample efficiency; asking a human thousands of comparison questions to learn a reward model (RM) is often not scalable.

A core problem of RLHF is active learning, where we want an agent to be judicious about the queries it asks to learn about a human's preferences as efficiently as possible (Sadigh et al., 2017; Casper et al., 2023; Baraka et al., 2025). Many active learning approaches require probabilistic modeling of uncertainty for computing data acquisition functions, making proper uncertainty representation an active area of research (Ovadia et al., 2019; Tran et al., 2020; Papamarkou et al., 2024). While Bayesian methods are well-principled, they are hard to scale to large neural networks (NN) (Izmailov et al., 2021). On the other hand, the simplicity of ensemble method (Dietterich, 2000; Lakshminarayanan et al., 2016) has made it a popular choice for active learning. However, training multiple models can be computationally intensive, especially for large NN reward models.

Due to recent advancements in approximate inference, Bayesian deep learning have become increasingly scalable (Daxberger et al., 2024; Shen et al., 2024). In this work, we develop a method called PreferenceEKF that enables efficient training of Bayesian neural networks for representing reward models in active preference-based reward learning. Specifically, by performing Bayesian filtering in a constructed neural network subspace, we maintain model uncertainty in a compute- and memory-efficient manner. The reduced dimensionality of the subspace enables application of

the extended Kalman filter, a classic inference method, for training neural networks. This allows sampling of arbitrary number of reward models from the model posterior, and use the samples for computing common uncertainty-based acquisition functions such as expected information gain and disagreement (Hennig & Schuler, 2012; Hernández-Lobato et al., 2014; Bıyık et al., 2022).

To the best of our knowledge, we are the first to leverage subspace filtering (Duran-Martin et al., 2022) to train neural network reward models for preference-based reward learning. We compare our method, PreferenceEKF to four widely used Bayesian deep learning methods for active preference-based reward learning, and conduct experiments in the D4RL (Fu et al., 2020) benchmark. Our findings are as follows:

- Active reward learning using PreferenceEKF leads to better sample efficiency (in terms of the number of queries required) compared to reward learning from random queries.
- PreferenceEKF performs on par with or better than all Bayesian deep learning baselines in terms of sample efficiency and calibration in preference modeling tasks.
- PreferenceEKF's runtime scales better with both model size and number of posterior samples used for computing acquisition function compared to all other methods.
- When the learned rewards are used for policy optimization in offline RL tasks (Levine et al., 2020), the reward model learned using active PreferenceEKF resulted in the best overall policy rollout performance compared to reward models learned using other methods.

## 2 RELATED WORK

**Reinforcement learning from human preferences.** While early works in reward learning focused on learning from expert demonstrations (Abbeel & Ng, 2004; Finn et al., 2016; Ho & Ermon, 2016), much recent interest has focused on reward learning from pairwise comparisons where human annotators are asked to compare two potential outcomes, e.g., labels, responses, or trajectories (Wirth et al., 2017; Christiano et al., 2017; Brown et al., 2019). Although preference feedback is much easier for annotators to provide than demonstrations, the minimal amount of information contained within a binary preference query necessitates collection of large amounts of feedback data.

Active learning is a widely used approach for minimizing the time-consuming process of collecting human feedback. It is a sequential problem in nature, as it iteratively collects the most useful data sample based on the model's current state, such as parameter posterior uncertainty. (Sadigh et al., 2017; Settles, 2009). While Bayesian methods have been successfully applied to obtain posteriors for active reward learning using lower-dimensional linear and Gaussian process reward models (Bıyık et al., 2022; 2024), it has not been widely adopted for neural reward models, since acquisition functions typically require sampling from the high-dimensional distribution of model parameters. Instead, ensembles have been the key enabler of neural network based active reward learning (Lee et al., 2021b; Christiano et al., 2017). Our work focuses on efficient yet performant posterior inference for active reward learning, without expensive training of multiple independent models.

**Uncertainty Quantification for neural networks.** Classic Bayesian methods that have been successfully used for neural network uncertainty quantification include Laplace approximation (Daxberger et al., 2024), Hamiltonian Monte Carlo (Neal, 2011), and variational inference (Blei et al., 2017). While not strictly motivated by Bayesian principles, the simplicity of ensemble method (Dietterich, 2000; Lakshminarayanan et al., 2016) and dropout (Srivastava et al., 2014; Gal & Ghahramani, 2016) has made them popular for UQ. While dropout method gets around ensemble method's expense cost of training multiple independent models, it been shown to lead to poor posterior approximation quality (Hron et al., 2018; Osband, 2016).

Bayesian filtering methods, which focuses on inferring hidden states from noisy observations, provide a principled approach to sequential learning, and have been widely used in robotics and signal processing (Thrun et al., 2005; Särkkä & Svensson, 2023). Application of Bayesian filtering for training neural networks (Singhal & Wu, 1988; de Freitas et al., 2000) has only recently been applied to deep neural networks via subspace methods by Duran-Martin et al. (2022) for neural bandits.

Instead of deriving epistemic uncertainty from posterior inference, a separate line of work has focused on leveraging nonparametric statistics techniques such as the bootstrap to perform UQ (Efron, 1992), and has successfully applied this technique for exploration in deep reinforcement learning (Osband et al., 2018; 2016). The same group of authors have also leveraged joint predictions for

UQ, and has applied the idea to finetuning large language models (Osband et al., 2023b;a). Our work leverages Bayesian filtering to train neural network reward models in active reward learning settings, where we focus primarily on parameter uncertainty instead of joint prediction uncertainty.

**Subspace methods for neural networks.** While there exists a vast literature on decreasing neural network size for efficient training and serving via architecture search (Elsken et al., 2019), quantization (Gholami et al., 2021), and pruning (Frankle & Carbin, 2022), we focus only on works that enable tractable inference in the reduced model. Specifically, there is growing evidence that the number of parameters required to fit a neural network is much smaller than its total parameter count; optimization and inference in the subspace spanned by these parameters offer not only computational efficiency, but also tractability of applying Bayesian methods for neural network training (Fort et al., 2020; Larsen et al., 2022). These parameters are found either as a subset of neural network parameters, or within a lower-dimensional subspace of the parameters.

Methods focusing on parameter subsets typically apply Bayesian methods such as Bayesian linear regression or variational inference to the last layer of the neural network, and point estimation methods like stochastic gradient descent (SGD) for intermediate layers (Snoek et al., 2015; Harrison et al., 2023; Brunzema et al., 2024). On the other hand, subspace methods typically constructs the low-dimensional subspaces via either random projection or singular value decomposition of SGD iterates of the full network; any inference or optimization technique such as sliced sampling (Izmailov et al., 2020) or SGD Li et al. (2018) can then be applied in the subspace in a tractable manner.

## 3 PRELIMINARIES

**Preference-based reward modeling.** We consider a Markov decision process (MDP) $\langle \mathcal{S}, \mathcal{A}, \mathcal{T}, r, \gamma \rangle$ with state space $\mathcal{S}$, action space $\mathcal{A}$, transition function $\mathcal{T}$, reward function $r : \mathcal{S} \to \mathbb{R}$, and discount factor $\gamma \in [0, 1)$. We assume access to a dataset of trajectories $\mathcal{D}^{traj} = \{\tau_1, \ldots, \tau_N\}$, where each trajectory $\tau_i$ is a sequence of $T$ steps $\tau_i = \{(s_{i,t}, a_{i,t}, s_{i,t+1})\}_{t=0}^{T-1}$, with each step consisting of state $s_t \in \mathcal{S}$, action $a_t \in \mathcal{A}$, and next-state $s_{t+1} \in \mathcal{S}$. In preference-based reward modeling, we do not assume access to a reward function. Instead, our task supervision comes from annotators who provide binary preference labels over pairwise trajectory comparisons queries, and the goal is to learn the annotator's reward function that informed their preference labels.

Formally, an annotator takes a trajectory pair query $Q_i = (\tau_a^i, \tau_b^i)$, and returns a preference label $y_i = \mathbb{1}(\tau_a^i \succ \tau_b^i) \in \{0, 1\}$ according to their internal reward function $r$. Given a dataset of queries and responses $\mathcal{D} = \{Q_i, y_i\}_i$, a widely-used approach for preference learning is to approximate $r$ with a parameterized reward model $r_{\boldsymbol{\theta}}$ via maximum likelihood estimation, where the likelihood $p_\theta(y \mid \tau_a, \tau_b)$ is typically defined using the Bradley-Terry (BT) model (Bradley & Terry, 1952),

$$p_\theta(y \mid \tau_a, \tau_b) = p_\theta(\tau_a \succ \tau_b) = \frac{\exp(\beta \cdot \mathcal{R}(\tau_a))}{\exp(\beta \cdot \mathcal{R}(\tau_a)) + \exp(\beta \cdot \mathcal{R}(\tau_b))} . \quad (1)$$

In particular, $\beta$ is a temperature parameter that models noisily optimal behavior of an annotator, and $\mathcal{R}(\tau_i)$ is the return of trajectory $\tau_i$ where the per-timestep reward is computed using a neural network-based RM $r_{\boldsymbol{\theta}}$, i.e., $\mathcal{R}(\tau_i) = \sum_{t=0}^{T-1} r_{\boldsymbol{\theta}}(s_{i,t})$ (Lee et al., 2021a). [1]

**Information-theoretic active learning.** We adopt the InfoGain acquisition function from Bıyık et al. (2022) for active preference-based reward learning, which assumes a distribution over RM parameters $p(\boldsymbol{\theta})$ such that, given a query-response pair $(Q, y)$ the predictive distribution is given by $p(y \mid Q) = \mathbb{E}_{p(\boldsymbol{\theta})}[p(y \mid Q, \boldsymbol{\theta})]$. It selects the query $Q_i$ that maximizes expected information gain on $\boldsymbol{\theta}$ by maximizing the mutual information between a query's answer label $y_i$ and $\boldsymbol{\theta}$:

$$Q_i^* = \arg\max_{Q_i} I\left(\boldsymbol{\theta}; y_i \mid Q_i, \boldsymbol{b}^{i-1}\right) \quad (2a)$$

$$= \arg\max_{Q_i} H\left(y_i \mid Q_i, \boldsymbol{b}^{i-1}\right) - \mathbb{E}_{\boldsymbol{\theta}}\left[H(y_i \mid \boldsymbol{\theta}, Q_i)\right] \quad (2b)$$

where $I$ is the mutual information, $H$ is the Shannon entropy (Cover & Thomas, 2006), and $\boldsymbol{b}^{i-1} = p(\boldsymbol{\theta} \mid \mathcal{D}_{1:i-1})$ is the posterior distribution over RM parameters after learning from $(i-1)$ queries.

---

[1]This formalism extends to state or state-action RMs, and whole trajectories or partial trajectory segments. Our experiments use state-based RM and partial trajectories.

We approximate this acquisition function via sampling as follows:

$$Q_i^* \doteq \arg\max_{Q_i} \frac{1}{M} \sum_{y_i \in \{0,1\}} \sum_{\boldsymbol{\theta} \in \boldsymbol{\Theta}} \left( P(y_i \mid Q_i, \boldsymbol{\theta}) \log_2 \left( \frac{M \cdot P(y_i \mid Q_i, \boldsymbol{\theta})}{\sum_{\boldsymbol{\theta}' \in \boldsymbol{\Theta}} P(y_i \mid Q_i, \boldsymbol{\theta}')} \right) \right) \qquad (3)$$

where $\boldsymbol{\Theta}$ is the set of models sampled from the posterior $\boldsymbol{b}^{i-1}$, and $M$ is the number of drawn samples. This approximation is asymptotically equivalent to Eq. 2b as $M \to \infty$. Due to the necessity of sampling models from the posterior $\boldsymbol{b}^{i-1}$, the work by Bıyık et al. (2020) has been limited to low-dimensional RMs, such as linear models. We now present our method, PreferenceEKF, which enables sampling of high-dimensional RMs, such as neural networks, that in turn allows us to scalably compute sampling-based acquisition functions like InfoGain to perform active learning.

## 4 METHOD

Sampling neural network models to approximate acquisition functions as in Eq. 3 can be expensive due to the high-dimensional parameter space of neural networks (Izmailov et al., 2021). Ensemble methods approximate this by training $M$ independent models, which can be infeasible for large $M$ and model sizes (Lakshminarayanan et al., 2016). We leverage the insight that neural networks are overparameterized and that solutions actually live in a much smaller subspace (Li et al., 2018), and perform posterior inference within this subspace. This allows us to sample an arbitrary number of models from a lower-dimensional posterior to approximate Eq. 3, without the overhead of training ensembles. We first show how to use extended Kalman filter (EKF), a widely used filtering algorithm, to train neural network reward models from preference data, then we show how to scale EKF to deep neural networks using subspace methods, as shown in Algorithm 1.

**Extended Kalman filter for training neural networks.** Using the formulation of sequential Bayesian inference, we perform posterior inference of neural network parameters from streaming data $\mathcal{D}_{1:i-1} = \{(Q_1, y_1), \ldots, (Q_{i-1}, y_{i-1})\}$. Starting from some prior belief $\boldsymbol{b}^0 = p(\boldsymbol{\theta})$ on the parameters, our posterior after observing $i$ samples can be expressed using Bayes' rule as follows:

$$p(\boldsymbol{\theta}_i \mid \mathcal{D}_{1:i}) \propto p(\mathcal{D}_i \mid \boldsymbol{\theta}_i) p(\boldsymbol{\theta}_i \mid \mathcal{D}_{1:i-1})$$

$$= \underbrace{p(\mathcal{D}_i \mid \boldsymbol{\theta}_i)}_{\text{Measurement}} \int \underbrace{p(\boldsymbol{\theta}_i \mid \boldsymbol{\theta}_{i-1})}_{\text{Dynamics}} \underbrace{p(\boldsymbol{\theta}_{i-1} \mid \mathcal{D}_{1:i-1})}_{\text{Previous posterior}} d\boldsymbol{\theta}_{i-1} \qquad (4)$$

where $p(\boldsymbol{\theta}_{i-1} \mid \mathcal{D}_{1:i-1})$ is the posterior belief over parameters after observing $i-1$ samples, which is combined with a parameter dynamics model and measurement model to form the posterior after observing the $i^{\text{th}}$ example. This formulation naturally allows for a recursive estimation scheme where model parameters can be updated by observing samples one at a time. To make computing Eq. 4 tractable, we assume additive Gaussian noise for both the dynamics model $p(\boldsymbol{\theta}_i \mid \boldsymbol{\theta}_{i-1}) = \mathcal{N}(\boldsymbol{\theta}_i \mid g(\boldsymbol{\theta}_{i-1}), \mathbf{U})$ and the measurement model $p(\mathcal{D}_i \mid \boldsymbol{\theta}_i) = \mathcal{N}(y_i \mid h(\boldsymbol{\theta}_i, Q_i), \mathbf{V})$, where $\mathbf{U} \in \mathbb{R}^{|\boldsymbol{\theta}| \times |\boldsymbol{\theta}|}$ and $\mathbf{V} \in \mathbb{R}^{|y| \times |y|}$ are prespecified Gaussian noise covariance matrices.

We treat neural network parameters as hidden states, and model the state dynamics $g : \mathbb{R}^{|\boldsymbol{\theta}|} \to \mathbb{R}^{|\boldsymbol{\theta}|}$ using an identity function. For preference learning, we model measurements $h : \mathbb{R}^{|\boldsymbol{\theta}|} \times \mathbb{R}^{|Q|} \to \mathbb{R}^{|y|}$ using BT model $p_\theta(\tau_a \succ \tau_b)$ computed using the learned RM $r_{\boldsymbol{\theta}}$ (Eq. 1). Assumptions on additive Gaussian noise and nonlinear dynamics and measurement functions make the neural network inference objective in Eq. 4 solvable in closed-form with the EKF algorithm, where the posterior takes a Gaussian form $\boldsymbol{b}^i = \mathcal{N}(\boldsymbol{\mu}_i, \boldsymbol{\Sigma}_i)$ where $\boldsymbol{\mu}_i \in \mathbb{R}^{|\boldsymbol{\theta}|}$ and $\boldsymbol{\Sigma}_i \in \mathbb{R}^{|\boldsymbol{\theta}| \times |\boldsymbol{\theta}|}$.

**Subspace inference.** Inference using EKF directly in the parameter space of a neural network is difficult, as the size of the covariance matrix $\boldsymbol{\Sigma}_i$ of the Gaussian posterior scales in $O(|\boldsymbol{\theta}|^2)$. We instead perform EKF in a learned subspace of the NN: we denote the full space parameter as $\boldsymbol{\theta}$ and subspace parameter as $\boldsymbol{z}$, where $|\boldsymbol{z}| \ll |\boldsymbol{\theta}|$, resulting in posterior $\boldsymbol{b}^i = \mathcal{N}(\boldsymbol{\mu}'_i, \boldsymbol{\Sigma}'_i)$ where $\boldsymbol{\mu}'_i \in \mathbb{R}^{|\boldsymbol{z}|}$ and $\boldsymbol{\Sigma}'_i \in \mathbb{R}^{|\boldsymbol{z}| \times |\boldsymbol{z}|}$. We further assume an affine mapping $\boldsymbol{\theta}(\boldsymbol{z}) = \mathbf{A}\boldsymbol{z} + \boldsymbol{\theta}_*$ that allows us to transform between the subspace and the full space. Here $\boldsymbol{\theta}_*$ is initialized via SGD on a small warm-up dataset in the full space. $\mathbf{A} \in \mathbb{R}^{|\boldsymbol{\theta}| \times |\boldsymbol{z}|}$ is a fixed projection matrix obtained from applying SVD to the SGD iterates ran in the full space, as shown on Line 8 through Line 10. [2]

---

[2]The projection matrix can also be computed via random projections (Li et al., 2018), but we found that the SVD approach (Izmailov et al., 2020) led to better empirical performance. See Fig. 3b for an ablation.

We perform EKF inference in the subspace to obtain an estimate $\boldsymbol{b}^i = p(\boldsymbol{z} \mid \mathcal{D}_{1:i})$ after observing each query-response pair $\mathcal{D}_i = (Q_i, y_i)$, then project each model sampled from $\boldsymbol{b}^i$ back to the full space via affine projection $\boldsymbol{\theta}(\boldsymbol{z})$ to perform the forward pass of the neural network to predict $\mathbb{1}(\tau_a^i \succ \tau_b^i)$. Predictive distribution for computing InfoGain is similarly computed via sampling followed by projection as $p(y \mid Q) = \mathbb{E}_{p(\boldsymbol{z})}[p(y \mid Q, \mathbf{A}\boldsymbol{z} + \boldsymbol{\theta}_*)]$. The belief update procedure is completely deterministic, with the only source of stochasticity coming from sampling of subspace parameters (followed by affine transformation) for computing the acquisition function and posterior predictive distribution.

**Active learning using subspace inference.** We refer to the ensemble-based approach as Deep-Ensemble and our approach as PreferenceEKF. We also assume a pool-based active learning setting (Settles, 2009) where we denote the set of all possible binary preference queries as $\mathcal{P}$. [3] For belief initialization (Line 12), whereas PreferenceEKF uses a zero-mean isotropic Gaussian of subspace dimension $|\boldsymbol{z}|$, DeepEnsemble initializes $M$ independent models each of dimension $|\boldsymbol{\theta}|$.

After belief initialization, the sequential phase of active learning begin. For random querying, Line 14 amounts to simply retrieving a random query from the query pool $\mathcal{P}$, whereas active learning algorithms computes an acquisition function for the optimal query to retrieve from the pool. The algorithm then receives the corresponding label for the retrieved query from an annotator in Line 15. For belief update (Line 16), whereas PreferenceEKF performs filtering in the constructed subspace only on the most recent query-response pair $\mathcal{D}_i$, DeepEnsemble trains each of the $M$ models using gradient descent on all data seen so far.

The most common uncertainty-based acquisition function is ensemble disagreement, i.e., pick the query $Q_i$ for which the predicted preference label $\mathbb{1}(\tau_a^i \succ \tau_b^i)$ has the highest variance across the ensemble. Disagreement has been popular for neural network-based active learning where it is expensive to scale Bayesian methods to high-dimensional settings (Christiano et al., 2017; Lee et al., 2021b), while InfoGain is the current state of the art acquisition function for lower-dimensional reward learning settings (Bıyık et al., 2020; 2024). While our method can be used to compute any sampling-based acquisition functions, we specifically leverage PreferenceEKF's ability to sample from high-dimensional distributions to scale InfoGain to neural network models. Due to the difficulty of sampling from high dimensional parameter distributions and the cost of training multiple models, DeepEnsemble approximates InfoGain by maintaining a small number of independent models. Dropout does so by sampling parameter dropout masks during inference.

---

**Algorithm 1** PreferenceEKF for active preference-based reward learning

---

1: **Input:**
2: $\mathcal{P}$: Set of all possible binary preference queries without labels
3: $\mathcal{D}^{\text{init}} = \{(Q_i, y_i)\}_{i=1}^{\tau}$: Initial preference dataset with $\tau$ query-label pairs
4: $B$: query budget limit
5: $w$: number of SGD iterations for subspace construction
6: **Procedure:**
7: # Subspace Construction
8: $\boldsymbol{\theta}_{1:w} = \text{SGD}(\mathcal{D}^{\text{init}})$
9: $\boldsymbol{\theta}_* = \boldsymbol{\theta}_w$          $\triangleright$ Parameter offset: $\boldsymbol{\theta}_* \in \mathbb{R}^{|\boldsymbol{\theta}|}$
10: $\mathbf{A} = \text{SVD}(\boldsymbol{\theta}_{1:w})$      $\triangleright$ Projection matrix: $\mathbf{A} \in \mathbb{R}^{|\boldsymbol{\theta}| \times |\boldsymbol{z}|}$
11: # Subspace Inference
12: $\boldsymbol{b}^0(\boldsymbol{z}) = \mathcal{N}(\boldsymbol{\mu}_0', \boldsymbol{\Sigma}_0')$      $\triangleright$ Belief initialization: $\boldsymbol{z} \in \mathbb{R}^{|\boldsymbol{z}|}$
13: **for** $t = 1 : B$ **do**
14:     $Q_t = \text{ComputeQuery}(\boldsymbol{b}^{t-1}, \mathbf{A}, \boldsymbol{\theta}_*, \mathcal{P})$    $\triangleright$ Compute InfoGain: $\boldsymbol{\theta}(\boldsymbol{z}) \in \mathbb{R}^{|\boldsymbol{\theta}|}$
15:     $y_t = \text{GetLabel}(Q_t)$
16:     $\boldsymbol{b}^t = \text{EKF}(\boldsymbol{b}^{t-1}, (Q_t, y_t))$         $\triangleright$ EKF update: $\boldsymbol{z} \in \mathbb{R}^{|\boldsymbol{z}|}$
17: **end for**

---

---
[3]Given a dataset of $N$ trajectories, there would be $|\mathcal{P}| = \binom{n}{2}$ possible pairwise comparison queries.

## 5 EXPERIMENTS

**Baselines and Evaluation.** We compare our PreferenceEKF method to four Bayesian deep learning baselines commonly used for reward modeling: DeepEnsemble, Dropout, Laplace, and Last-Layer MCMC (LLMCMC), which we detail in Section A.2.1. We address the following questions: (1) Does active learning with PreferenceEKF lead to more data-efficient and effective preference-based reward learning compared to the baselines (2) Can the reward models sampled from PreferenceEKF's induced posterior be used to for policy optimization via offline RL? (3) Does representing parameter uncertainty $p(\boldsymbol{\theta} \mid \mathcal{D})$ as a subspace distribution lead to computational advantages over representation using ensembles and dropout? We additionally study the model calibration capability of all algorithms, as well as ablate the subspace construction procedure for PreferenceEKF.

In the reward learning experiments, given a limited query budget $B$, we would like to learn RMs from preference queries as sample-efficiently as possible. Evaluation is done by comparing the Bradley-Terry log-likelihood achieved by a RM on a held-out set of test queries throughout training. To create the preference query pool $\mathcal{P}$, we randomly sample pairwise partial trajectories from a trajectory dataset $\mathcal{D}^{traj}$, then generate noisily optimal synthetic labels: for a given pair of trajectories, we compute their returns and sample a preference label according to the BT model (Eq. 1).

In the offline RL experiments, The learned RMs are then used for training parameterized policies $\pi_\phi(a \mid s)$ via offline RL. This is done by first labeling the trajectory dataset $\mathcal{D}^{traj}$ with the learned RM: we take the average predicted reward over $M$ models $r_{\boldsymbol{\theta}}^M(s_{i,t}) = \frac{1}{M}\sum_{m=1}^M r_{\boldsymbol{\theta}}^m(s_{i,t})$ for each state, where $r_{\boldsymbol{\theta}}^m$ is the $m^{\text{th}}$ sampled reward model for PreferenceEKF and Dropout, and the $m^{\text{th}}$ model in the ensemble for DeepEnsemble. A reward-labeled trajectory takes the form, $\tau_i = \{(s_{i,t}, a_{i,t}, s_{i,t+1}, r_{\boldsymbol{\theta}}^M(s_{i,t}))\}_{t=0}^{T-1}$, and we train policies on the reward-labeled $\mathcal{D}^{traj}$ using Implicit Q-Learning (IQL) (Kostrikov et al., 2021), an empirically successful offline RL algorithm. We evaluate policies by comparing their empirical rollout returns throughout RL training.

**Tasks.** We evaluate our approach in D4RL (Fu et al., 2020), a popular offline RL benchmark, and choose a mixture of environments spanning MuJoCo locomotion (HalfCheetah, Hopper, Walker2d), Adroit Shadow Hand (pen twirling), and Maze2D navigation. Within each environment, we choose trajectory datasets of varying characteristics: MuJoCo trajectories span a range of performance quality, Adroit trajectories are generated by a human operator and a fine-tuned expert-level RL policy, and maze navigation trajectories are collected from policies executed in mazes of varying difficulty. We consider each dataset as a separate task, for a total of 12 tasks. While our main experiments focus on simulated state-based control tasks, we refer to Appendix A.2.3 for results on pixel-based tasks.

**Implementation Details.** Unless otherwise stated, all experiments are done on a single node with 8 NVIDIA RTX A6000 GPUs via sharding, query budget $B = 60$, and trajectory segments of length 50. On the belief update step (Line 16), PreferenceEKF learn from only the most recent query-label pair, while all baselines learn from all data seen so far. Further DeepEnsemble is the only method that needs to train multiple models, so we set $M = 5$ as is commonly done for ensemble-based uncertainty quantification (Ovadia et al., 2019); all other methods can sample arbitrary number $M$ of models from the learned posterior, so we set $M = 100$ for them. With the exception of the scaling experiments in Section 5.3 and the ablation experiments in Section 5.5, all reward models are represented as multi-layer perceptrons (MLP) with two hidden layers of 64 units, using subspace dimensionality $|z| = 200$. All methods use the InfoGain acquisition function to ensure fair comparison. We show additional results using the disagreement acquisition function in Appendix Section A.2.2, and Appendix Section A.2.1 for more details on baseline implementations.

### 5.1 DOES PREFERENCEEKF LEAD TO SAMPLE-EFFICIENT ACTIVE REWARD LEARNING?

Given a fixed query budget per task, we evaluate each algorithm over 5 seeds. We use state-based partial trajectories, and compute return of each trajectory as $\mathcal{R}(\tau_i) = \sum_{t=1}^T r_{\boldsymbol{\theta}}(s_{i,t})/T$. We show in Fig. 1a that aggregated over all tasks, active PreferenceEKF achieves higher sample efficiency compared to its random counterpart. Additionally, both random and active variants of PreferenceEKF performs on par with or outperforms all other baselines. In the appendix, we show in Fig. A.1 that in most task, active PreferenceEKF outperforms both its random counterpart as well as all other

algorithms in terms of sample efficiency and final log-likelihood. See Appendix Section A.1 for statistical testing results backing up these empirical observations.

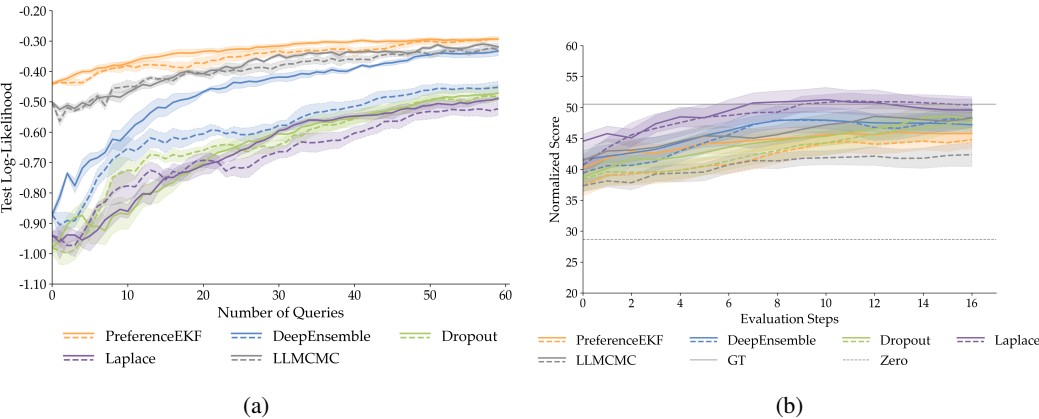

(a)                                                    (b)

Figure 1: Fig. 1a shows comparison of the random (dashed line) and active (solid line) variants of each algorithm for preference-based reward modeling using the InfoGain acquisition function, aggregated over 12 D4RL tasks (mean±s.e. over 5 seeds). Fig. 1b shows comparison of policy optimization using the reward models learned from random and active variants of each algorithm, aggregated across 12 D4RL tasks in the offline RL setting (mean±s.e. over 5 seeds). See Fig. A.1 and Fig. A.8 in the appendix for per-task results for reward learning and offline RL evaluations results, respectively. All results here are shown with a moving average over the last 5 evaluations.

### 5.2 CAN RMS LEARNED USING PREFERENCEEKF BE USED FOR POLICY OPTIMIZATION?

The goal of the offline RL experiments is to test whether an RM learned from limited number of preference queries can recover the ground-truth reward information, evaluated by whether the learned RM can induce a policy that reaches or exceeds the performance of a policy trained with ground-truth environment rewards (GT), and whether the resulting policy can outperform a separate RL policy trained on zerod out rewards (Zero). All policies are trained using IQL (Kostrikov et al., 2021) over 5 seeds on the reward-labeled dataset for 1M steps. Evaluation is done via 5 rollouts every 50K steps. In Fig. 1b, we show that when aggregated across all tasks, reward models learned from all methods converge to similar policy performance, with all methods performing on par with or slightly worse than the GT policy, and all methods greatly outperforming the Zero policy.

While policy optimization using reward models learned from active variant of PreferenceEKF slightly outperforms that learned from random PreferenceEKF, we note both variants reached roughly the same final log-likelihood in Fig. 1a, leading to the observed similar policy optimization performance in Fig. 1b.

### 5.3 HOW DOES MODEL TRAINING WITH PREFERENCEEKF SCALE?

Next, we investigate whether extended Kalman filter can serve as a scalable alternative to gradient descent for training neural network reward models using preference data. Computing predictive distributions via sampling (Eq. 3) requires forward passes over $M$ neural networks. We show here the computational advantage of PreferenceEKF in maintaining a subspace parameter posterior distribution $p(\boldsymbol{\theta} \mid \mathcal{D})$ to sample models from, compared to DeepEnsemble's approach of maintaining and training $M$ neural networks explicitly and Dropout's approach of sampling dropout masks. We run all scaling experiments on CPUs as the larger models and ensemble sizes led to out-of-memory errors. Finally, while PreferenceEKF's belief update procedure (Line 16) only requires the most recent query due to EKF being an online estimation algorithm, DeepEnsemble and Dropout train over all queries observed so far (Lee et al., 2021b; Christiano et al., 2017). All experiments use subspaces of fixed dimensionality $|\boldsymbol{z}| = 200$.

We show in Fig. 2a that given a fixed architecture of a two-layer MLP with 64 units per layer, the runtime of PreferenceEKF for learning a reward model from $B = 60$ queries scales much

more gracefully with increasing $M$ compared to DeepEnsemble. While Dropout does not need to maintain multiple independent models, it is still slower than PreferenceEKF as it performs model update in full parameter space instead of a lower-dimensional subspace. Fig. 2b demonstrates that final test log-likelihood favors PreferenceEKF over the other methods, showcasing that our approach maintains consistent performance on top of computational efficiency given increasing $M$.

Fig. 2c and Fig. 2d show that given fixed number of model parameter samples ($M = 5$) and increasing neural network architecture size, PreferenceEKF scales more gracefully compared to other methods, on top of maintaining test log-likelihood performance. This showcases the scalability of subspace training to not only settings where we need large number of model samples $M$, but also to settings where we need larger neural networks $|\boldsymbol{\theta}|$.

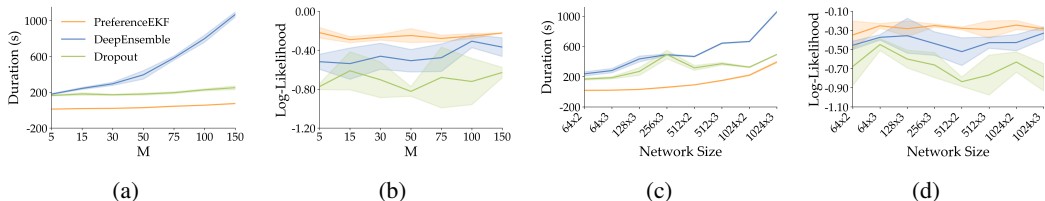

(a)            (b)            (c)            (d)

Figure 2: Fig. 2a and Fig. 2b show the effect of scaling the number of model samples $M$, while Fig. 2c and Fig. 2d show the effect of scaling neural network architecture size in the active learning setting (mean $\pm$ std over 3 seeds). Overall, PreferenceEKF scales more gracefully than the other algorithms, showcasing the advantages of both subspace training and uncertainty representation using subspace distribution over model ensembles and dropout masks.

## 5.4 DOES PREFERENCEEKF LEAD TO BETTER CALIBRATED MODELS?

While effective representation of parameter uncertainty is crucial for efficient active learning, it is also important for calibration of neural network predictions (Guo et al., 2017; Ovadia et al., 2019). We study whether uncertainty quantification (UQ) using subspace methods leads to better calibrated model predictions compared to UQ using ensemble methods and dropout, as quantified by two commonly used UQ metrics: expected calibration error (ECE) using 5 bins (Naeini et al., 2015; Pavlovic, 2025) and Brier score (Brier, 1950; DeGroot & Fienberg, 1983). We provide further calibration experiment details in Appendix Section A.2.4.

We show in Fig. 3a that variants of PreferenceEKF has the lowest ECE among all methods, and the second lowest Brier score along with active DeepEnsemble. This highlights the quality of posterior approximation achieved by subspace inference methods. While dropout-based methods gets around the computational cost of ensembles, the resulting uncertainty representation has led to both poorer active learning and UQ results as compared to subspace methods.

## 5.5 ABLATION STUDY ON SUBSPACE CONSTRUCTION

The method for subspace construction for PreferenceEKF can be modified to 1) use varying dimensionality of the subspace, and to 2) use random projection to generate the subspace basis instead of running SVD on gradient descent iterates (Li et al., 2018; Izmailov et al., 2020). While all of our experiments so far use a fixed dimensionality of $|\boldsymbol{z}| = 200$ with SVD-based construction, we perform an ablation analysis over these choices, as shown in Fig. 3b. We observed that while the SVD-based approach works well for smaller subspace dimensions, the random projection approach can eventually reach performance on par with or even outperform the SVD approach as the subspace dimension increases. This finding is similar to what was observed by Duran-Martin et al. (2022) in bandit settings, highlighting the generality of this result for subspace Bayesian filtering methods used to train neural networks.

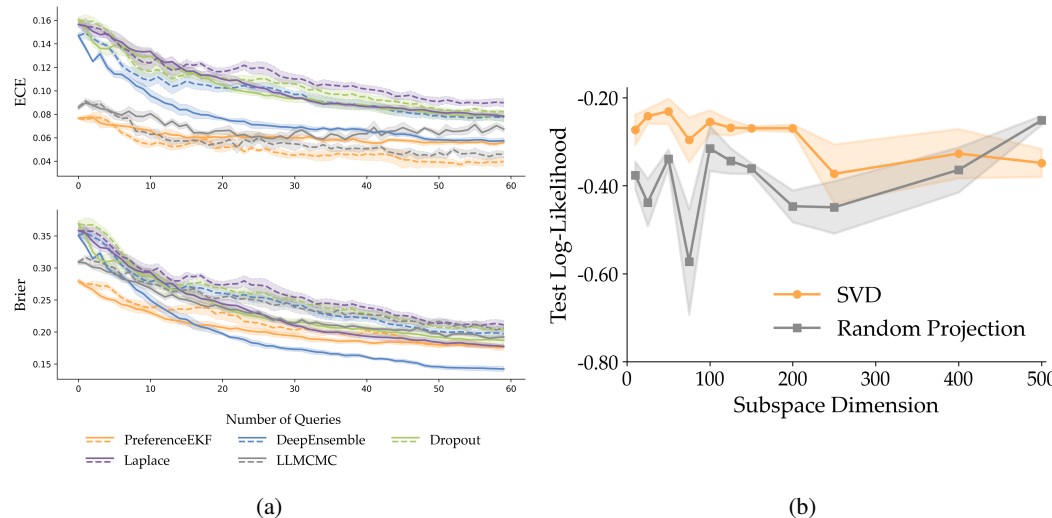

(a)                                                                (b)

Figure 3: Fig. 3a shows comparison of the random (dashed line) and active (solid line) variants of the algorithms in model calibration, as evaluated by expected calibration error and Brier score on a test dataset (lower is better for both metrics). Fig. 3b shows an ablation over the subspace construction technique for PreferenceEKF, as evaluated by log-likelihood on a test dataset (higher is better). Both the UQ experiment and ablation analysis here are performed over 3 seeds (mean ± std) on the Walker Medium Expert task.

## 6 CONCLUSION

In this work, we successfully adopted extended Kalman filters to train neural networks in active preference-based reward modeling setting. We showed several advantages of maintaining a subspace distribution over neural network parameters $p(\boldsymbol{\theta} \mid \mathcal{D})$, in comparison to four other widely used Bayesian deep learning methods for active reward learning. Our approach led to more sample efficient active reward learning, similarly performant RL policy optimization, better runtime scaling with respect to model size and model sample count, and better calibration through higher-quality uncertainty representation. Learning a lower-dimensional distribution of neural network parameters further allowed us to scale the current state of the art acquisition function for preference-based active reward modeling, InfoGain (Bıyık et al., 2020), from lower-dimensional model settings to deep neural networks.

**Limitations and future work.** While we found subspace method to be an effective tool for scaling Bayesian filtering methods for neural network training, it is unsure whether this approach will be effective for applying Bayesian methods to foundation model-scale reward models (Mahan et al., 2024; Zhang et al., 2024). Due to the unimodality of the Gaussian distribution that extended Kalman filter maintains, alternative methods may need to be investigated for approximating multimodal posteriors, e.g., learning reward functions from annotators with differing preferences (Poddar et al., 2024; Siththaranjan et al., 2023). We would further like to evaluate uncertainty quantification using the recent works on epistemic neural networks (Osband et al., 2023b), which focuses on joint predictions uncertainty instead of marginal predictive distribution.

While our work primarily focused on improving sample-efficiency of reward modeling in RLHF, we would like to further investigate how learned posterior distribution of reward models can aid in RL policy's exploration and serve as a mechanism for mitigating reward hacking (Yang et al., 2024; Gao et al., 2022; Hadfield-Menell et al., 2017). Finally, due to its sample-efficiency and adaptivity to non-stationary distributions, we believe the subspace filtering method to be a viable candidate for uncertainty quantification and large model finetuning in robot learning domains (Bellemare et al., 2017; Fridovich-Keil et al., 2020; Bobu et al., 2020).

## 7 REPRODUCIBILITY STATEMENT

Our code is anonymously available in the JAX framework at `https://github.com/preferenceEKF2025/preference_ekf`. We ensured that all pseudo-randomness has been controlled for via JAX's PRNG implementation. We provide all SLURM launch scripts, visualization scripts, and configuration files with all hyperparameters as part of code release.

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

# A  TECHNICAL APPENDICES AND SUPPLEMENTARY MATERIAL

Our code is available in the JAX (Bradbury et al., 2018) framework at `https://github.com/preferenceEKF2025/preference_ekf`. For implementation of the reward learning algorithms, we use Dynamax (Linderman et al., 2025) for extended Kalman filtering (EKF), Laplax (Weber et al., 2025) for Laplace approximation, and Blackjax (Cabezas et al., 2024) for MCMC. For offline RL, we use Unifloral (Jackson et al., 2025) for implementation of implicit Q-learning (IQL). All statistical testing are done using SciPy (Virtanen et al., 2020). Unless stated otherwise, all experiments are done on a single node with 8 NVIDIA RTX A6000 GPUs via SLURM sharding.

## A.1  STATISTICAL TESTING

To provide statistical significance to the main claims from Section 5.1, we conduct hypothesis testing of 1) whether the active variant of each algorithm outperforms its random variant and 2) whether active PreferenceEKF outperforms active variants of other Bayesian deep learning baselines. For the summary statistic of each active reward learning experiment run, we compute the normalized area under curve (AUC) of the log-likelihood plot in Fig. 1a. This measures the rate of improvement for log-likelihood. Since all runs are performed using the same set of 5 random seeds and the same train/test dataset split, we conduct our hypothesis testing using the one-sided paired $t$-test to compare the normalized AUC between two sets of runs. We additionally compute the $95\%$ confidence interval as well as Cohen's $d$ for effect size.

In the first 5 rows of Table 1, we show the performance of active versus random variant of each algorithm, where each row is conducted over 5 seeds. We see that active PreferenceEKF and DeepEnsemble outperforms their random counterparts in normalized AUC with high statistical significance, while Dropout, Laplace and LLMCMC fail to do so. We also note that DeepEnsemble and LLMCMC reach roughly the same log-likelihood results as PreferenceEKF.

In the last 4 rows of Table 1, we show the performance of active PreferenceEKF versus active variant of other baselines, where each row is conducted over 5 seeds. We see that active PreferenceEKF outperforms active variants of all baselines in normalized AUC with high statistical significance.

| Test | $t$ | $p$-value | Cohen's $d$ | 95% CI |
|---|---|---|---|---|
| EKF (A vs. R) | 2.43 | 0.036 | 1.01 (large) | $(0.00, \infty)$ |
| Ensemble (A vs. R) | 15.08 | $< 0.001$ | 5.21 (large) | $(0.09, \infty)$ |
| Dropout (A vs. R) | -0.69 | 0.737 | -0.44 (small) | $(-0.05, \infty)$ |
| Laplace (A vs. R) | 0.82 | 0.230 | 0.47 (small) | $(-0.02, \infty)$ |
| LLMCMC (A vs. R) | 1.46 | 0.109 | 0.67 (medium) | $(-0.00, \infty)$ |
| EKF vs. Ensemble | 27.44 | $< 0.001$ | 7.80 (large) | $(0.11, \infty)$ |
| EKF vs. Dropout | 16.77 | $< 0.001$ | 11.22 (large) | $(0.25, \infty)$ |
| EKF vs. Laplace | 19.44 | $< 0.001$ | 12.55 (large) | $(0.26, \infty)$ |
| EKF vs. LLMCMC | 5.21 | 0.003 | 3.84 (large) | $(0.03, \infty)$ |

Table 1: One-sided paired $t$-tests comparing active vs. random variants of each algorithm, and active EKF vs. active variant of other baseline algorithms.

## A.2  PREFERENCE-BASED REWARD LEARNING

**Implementation Details.** Unless otherwise stated, all reward learning experiments are done using subspace dimensionality $|z| = 200$, query budget $B = 60$, and partial trajectory of length 50. All neural networks reward model are represented using multi-layer perceptrons (MLP) with two hidden layers of 64 units. We apply normalization to all input features. PreferenceEKF and Dropout uses $M = 100$ model parameter samples to compute the acquisition function and posterior predictive distribution, while DeepEnsemble trains $M = 5$ independent networks, each with different weight initialization and randomness for minibatch shuffling.

All tasks use a pool of 150K pairwise partial trajectory queries drawn from the trajectory dataset to perform random or active querying over, and 3000 test queries for log-likelihood evaluation.

For generation of noisy-optimal synthetic labels, we apply trajectory return normalization before passing trajectory pairs through the BT model (Eq. 1) to compute the likelihood $p_\theta(\tau_a \succ \tau_b)$. We use temperature parameter of $\beta = 7$, resulting in roughly 5-15% mistaken preference labels per task.

Before the sequential learning phase starting on Line 13, all algorithms receive a small dataset consisting of $\tau = 8$ query-response pairs for belief initialization, i.e., all algorithms observe a total of $\tau + B = 8 + 60 = 68$ samples. All algorithms run variants of gradient descent (GD) on the warm-up dataset for 420 optimizer steps. While PreferenceEKF uses SGD with learning rate of 1e-4, momentum of 0.9, and batch size of 1, DeepEnsemble and Dropout uses Adam (Kingma & Ba, 2014) with learning rate of 1e-4 along with default hyperparamters from Optax (DeepMind et al., 2020), and batch size of 8. Previous works have found that SGD with a high constant learning is crucial to producing GD iterates with enough variance to construct a subspace effective for optimization and inference (Fort et al., 2020), hence the different choice of optimizer for PreferenceEKF.

PreferenceEKF constructs the subspace by running SVD on the GD iterates obtained from running SGD on the warmup dataset. We throw away the first 20 out of the 420 GD iterates and keep only every other remaining iterate, for a total of $(420 - 20)/2 = 200$ iterates. Thus, SVD takes in a model parameter array of shape $(200 \times |\boldsymbol{\theta}|)$, and return a projection matrix $\mathbf{A}$ of shape $(200 \times |\boldsymbol{z}|)$ by keeping only the top $|\boldsymbol{z}| = 200$ principal components. The final GD iterate is used as the full space parameter offset $\boldsymbol{\theta}_*$, which, along with projection matrix $\mathbf{A}$, is used to transform from the subspace back up to the full space for, e.g. computing predictive distributions as described in Section 4. Finally, PreferenceEKF performs belief initialization (Line 12) in the subspace using a zero-mean isotropic Gaussian of dimension $|\boldsymbol{z}| = 200$.

On the belief update step (Line 16), PreferenceEKF learns from only the most recent query-label pair, while DeepEnsemble and Dropout learns learns from all data seen so far over 3 epochs. Note that the specific filtering algorithm we use is the iterated EKF (Bell & Cathey, 1993), which repeatedly re-linearize the measurement model around the estimated posterior. Empirically, we observed better log-likelihood evaluation performance in exchange for marginally extra runtime. We refer to the number of such re-linearization steps on every new sample as $n_{\text{linearize}}$. For further details on iterated EKF, refer to Section 8.3.2.2 of (Murphy, 2023). We use $n_{\text{linearize}} = 5$, prior noise of 0.07, systems noise of 1e-3, and measurement noise of 0.07 for all of our PreferenceEKF experiments.

### A.2.1 BASELINE ALGORITHMS

The primary tradeoff that Bayesian deep learning (BDL) algorithms are concerned with is the computational tractability and approximation quality of the posterior distribution over model parameters given data $p(\boldsymbol{\theta} \mid \mathcal{D})$. For high-dimensional models such as neural networks, the posterior can be highly multi-modal, which can be difficult to approximate for algorithms that use unimodal distributions (typically Gaussian) such as Laplace approximation and extended Kalman filters. On the other hand, while Markov chain Monte Carlo (MCMC) has been the gold standard for posterior approximation (Izmailov et al., 2021), they are very difficult to scale to large models with many parameters. As such, many BDL algorithms try to "be Bayesian" over only a subset or subspace of model parameters, or rely on ensembling to hopefully reach multiple posterior modes. Here we provide a high-level description of the five classes of BDL algorithms we use for our experiments, the corresponding implementation details, as well as where they have been used in the reward learning literature.

**DeepEnsemble** and **Dropout** are among the most widely-used BDL algorithms for reward modeling and more generally, uncertainty quantification in neural networks (Christiano et al., 2017; Gleave & Irving, 2022; Chen et al., 2020; Hoque et al., 2022). They approximate the posterior by relying on randomness (e.g., weight initialization, mini-batch sampling order) to train multiple models and average over their predictions. While DeepEnsemble has the computational burden of actually training multiple neural networks, Dropout masks out subset of model parameters during training and computes the posterior predictive distribution by averaging predictions from multiple model copies with different weight masks during inference time, thus requiring training of only one model. The idea for both approach is for the multiple resulting models to act as samples from the posterior distribution. All $M$ models trained under DeepEnsemble method receive a different stream of mini-batches for training. Dropout uses weight dropout probability of 0.3 for all experiments, during both training and inference.

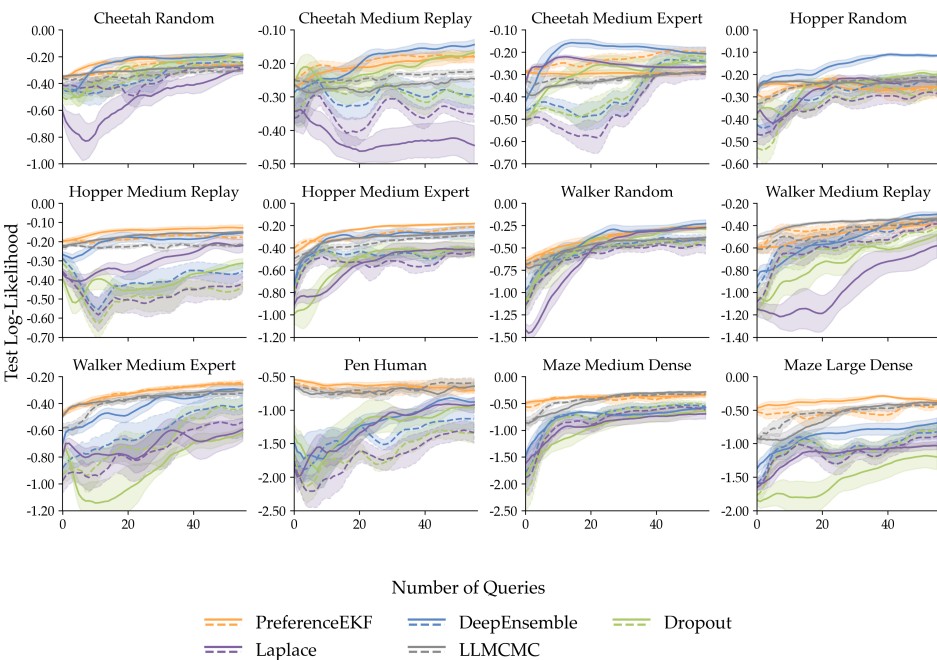

Figure A.1: Comparison of the random (dashed line) and active (solid line) variants of the algorithms using the InfoGain acquisition function, across 12 D4RL tasks for preference-based reward modeling (mean±s.e. over 5 seeds). In all tasks, active PreferenceEKF either performs on par with or outperforms other algorithms in terms of sample-efficiency and final log-likelihood.

**LLMCMC**: Despite the high quality posterior approximation of MCMC methods for smaller models such as linear models (Bıyık et al., 2020; Hadfield-Menell et al., 2017), they have are not widely used for neural network posterior inference due to their poor scalability to parameter count. Most application of MCMC to BDL trains the entire NN model using more efficient maximum likelihood methods like gradient descent, then perform MCMC only over the parameters of the final layer. We chose this "last-layer Bayesian" approach as it has been shown to strike a good balance between computational tractability and approximation quality (Brown et al., 2020; Snoek et al., 2015). The specific MCMC sampler we use is NUTS (Hoffman & Gelman, 2014). On active learning step, we construct a new log-density function using the aggregated dataset using all samples seen so far. For belief initialization, we use 500 warm up MCMC iterations followed by 1000 additional iterations. For belief update steps, since the log-density function should not differ too much with one additional aggregated sample, we set warm up iteration to be 20, followed by 1000 additional iterations. We then subsample $M$ models from the resulting MCMC iterates to form our sampling-based posterior.

**Laplace**: While Laplace approximation (LA) has traditionally been used for smaller models in logistic regression and Gaussian process-based regression models (Biyik et al., 2020; Rasmussen & Williams, 2005), recent advancements such as those in Dangel et al. (2025); Daxberger et al. (2024) have made the technique highly scalable to neural network architectures. Combined with parameter efficient finetuning technique such as LoRa (Hu et al., 2021), LA has even been applied to transformer-scaled reward models (Yang et al., 2024). By approximating likelihood curvature around a model solution trained via maximum likelihood methods such as gradient descent, LA constructs a local Gaussian approximation to the model posterior. We use the full curvature approximation-based approach of Weber et al. (2025) to perform LA over the entire reward model, with prior precision value of 1000. Once the curvature information has been constructed for the Gaussian posterior approximation, we can sample arbitrary number of model parameters from the posterior.

**PreferenceEKF**: While the preceding described methods perform inference over either the full model or a subset thereof, PreferenceEKF find a low dimensional subspace (as opposed to just a subset of the parameters) within the full parameter space, and perform inference within the sub-

space. The main insight of subspace inference approaches (Daxberger et al., 2021) is that due to the overparameterized nature of neural networks, capturing posterior information only over a constrained subspace would be a sufficient alternative to posterior inference over the whole network. Once a Gaussian approximation is obtained via subspace Kalman filtering, we can sample arbitrary number of model parameters from the posterior.

### A.2.2 ACQUISITION FUNCTIONS

The InfoGain acquisition function introduced in Eq. 2a was developed by Bıyık et al. (2020) for active reward learning using linear reward models. To motivate its origin, we first express the Info-Gain objective in three equivalent forms below due to symmetry of mutual information. We refer to Section 5 of Bıyık et al. (2020) for further interpretations of the objective, and Appendix 9.1 of their work for derivation of the sampling-based approximation shown in Eq. 3.

$$Q_i^* = \arg\max_{Q_i} I\left(\boldsymbol{\theta}; y_i \mid Q_i, \boldsymbol{b}^{i-1}\right) \tag{5a}$$

$$= \arg\max_{Q_i} H\left(\boldsymbol{\theta} \mid Q_i, \boldsymbol{b}^{i-1}\right) - \mathbb{E}_{y_i}\left[H(\boldsymbol{\theta} \mid y_i, Q_i, \boldsymbol{b}^{i-1})\right] \tag{5b}$$

$$= \arg\max_{Q_i} H\left(y_i \mid Q_i, \boldsymbol{b}^{i-1}\right) - \mathbb{E}_{\boldsymbol{\theta}}\left[H(y_i \mid \boldsymbol{\theta}, Q_i)\right] \tag{5c}$$

The idea of mutual information-based acquisition functions is rooted in the concept of expected information gain studied in Bayesian optimal experiment design and active data selection (MacKay, 1992; Lindley, 1956). It was later extended to Bayesian optimization using Gaussian process models under the methods Bayesian active learning by disagreement (BALD) (Houlsby et al., 2011), entropy search (ES) (Hennig & Schuler, 2012), and predictive entropy search (PES) (Hernández-Lobato et al., 2014). In particular, the mutual information objective function in Eq. 5a is expressed in its ES form in Eq. 5b, and expressed in its equivalent but computationally efficient PES form in Eq. 5c. Our PreferenceEKF method focuses on efficient sampling of high-dimensional neural network model parameters to approximate the predictive distribution for optimizing Eq. 5c.

Although our main experiments all use the InfoGain acquisition function to showcase the advantage of being able to sample from high-dimensional neural network parameter distributions, the PreferenceEKF method is agnostic to the acquisition function used for active learning. While Fig. 1a and Fig. A.1 showcase the aggregate and per-task log-likelihood results for active preference-based reward learning experiments using InfoGain, here we show additional results using the widely used disagreement acquisition function, which selects the query $Q_i$ for which the predicted preference label $\mathbb{1}(\tau_a^i \succ \tau_b^i)$ has the highest variance across the ensemble or sampled models. Fig. A.2 and Fig. A.3 show the aggregate and per-task log-likelihood results, while Fig. A.4 show the calibration results. Overall, we see that InfoGain led to superior active reward learning performance compared to disagreement.

### A.2.3 PIXEL-BASED REWARD LEARNING

While our main results in Section 5 are performed on state-based control tasks, here we showcase the applicability of PreferenceEKF to pixel-based tasks. We focus on the Visual D4RL (V-D4RL) benchmark (Lu et al., 2023), which contains rendered pixel-image observations corresponding to datasets from the state-based D4RL benchmark.

Our pixel-based reward model architecture consists of an ImageNet-pretrained ResNet18 image encoder (Deng et al., 2009; He et al., 2016) as the backbone and a two-layer MLP with 256 hidden units per layer as the reward prediction head. We finetune the entire reward model via SGD as part of the belief initialization step of Line 12, and perform EKF inference within the subspace of only the reward head parameters while keeping the finetuned backbone frozen. Due to the increased task and model complexity, we construct a subspace with dimensionality of 500 (compared to 200 in the state-based tasks with smaller reward models), and use random projection to do so since a larger subspace benefits equally from random projection versus SVD-based construction techniques as shown in Fig. 3b.

Since EKF's belief update procedure scales cubically with dimensionality of the observation space, we use a measurement likelihood function (Eq. 4) over trajectory embeddings rather than raw trajectory pixels. We compute embeddings from the final layer of the ResNet18 backbone before the

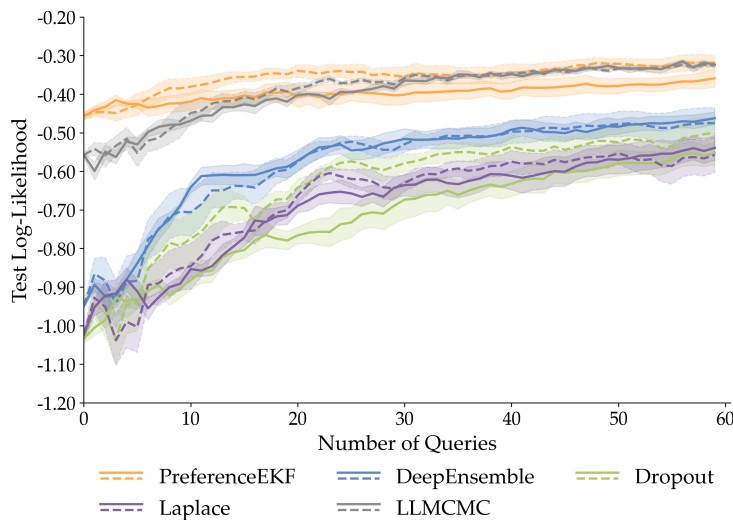

Figure A.2: Comparison of the random (dashed line) and active (solid line) variants of the algorithms for preference-based reward modeling using the disagreement acquisition function, aggregated over 12 D4RL tasks (mean±s.e. over 5 seeds). While PreferenceEKF and LLMCMC outperforms all other methods, their active variants did not outperform their random variants.

reward prediction head, and mean-pool the embeddings across all timesteps of a trajectory segment to obtain embeddings that aggregate trajectory-level information. Empirically, raw pixel observations over trajectory segment lengths of 10 steps with images of height, width, channel $(84, 84, 3)$ would result in observation dimension of $10 \times 84 \times 84 \times 3 = 211,680$ per trajectory, while mean-pooled embedding-based observation results in dimension of $512$ per trajectory.

To finetune the pixel-based reward model, we start with a much bigger initial query dataset of $150$ (compared to just 8 in state-based experiments), and use learning rate of $0.0001$ over 3000 mini-batches with batch size 16. In Fig. A.5 and Fig. A.6, we show that PreferenceEKF is indeed a viable method for active preference-based reward learning. While the performance of active versus random sampling varies heavily across the three chosen pixel-based tasks, the active variant as a whole shows promising improvement over the random variant. We leave research on EKF variants that efficiently scale with observation dimension, and more parameter efficient subspace inference methods such as those based on LoRa (Hu et al., 2021) to future work.

### A.2.4 Model calibration experiments

In addition to the results from Section 5.4 on expected calibration error and Brier scores, we provide in Fig. A.7 reliability diagrams computed from model predictions over all tasks and seeds. Due to the per-timestep parameterization of the reward model for computing the Bradley-Terry loss function Eq. 1, our binary preference query dataset is implemented to always have the second item be preferred over the first item. This corresponds to label of always 1, hence why the reliability diagrams only show calibration for half of the probability line. Upon inspection, we can see that PreferenceEKF and LLMCMC exhibit the lowest model calibration error.

### A.3 Offline reinforcement learning

The extent to which offline RL algorithms leverages reward information for policy optimization, i.e., whether reward-induced policy performance is a good metric for assessing learned reward models, is heavily dependent on the trajectory dataset: when ran on datasets consisting solely of expert demonstrations, offline RL algorithms will largely ignore reward information and adopt a behavioral cloning-like learning strategy. On the other hand, it is generally difficult to train a policy from a dataset consisting of purely random behavior (Kumar et al., 2021).

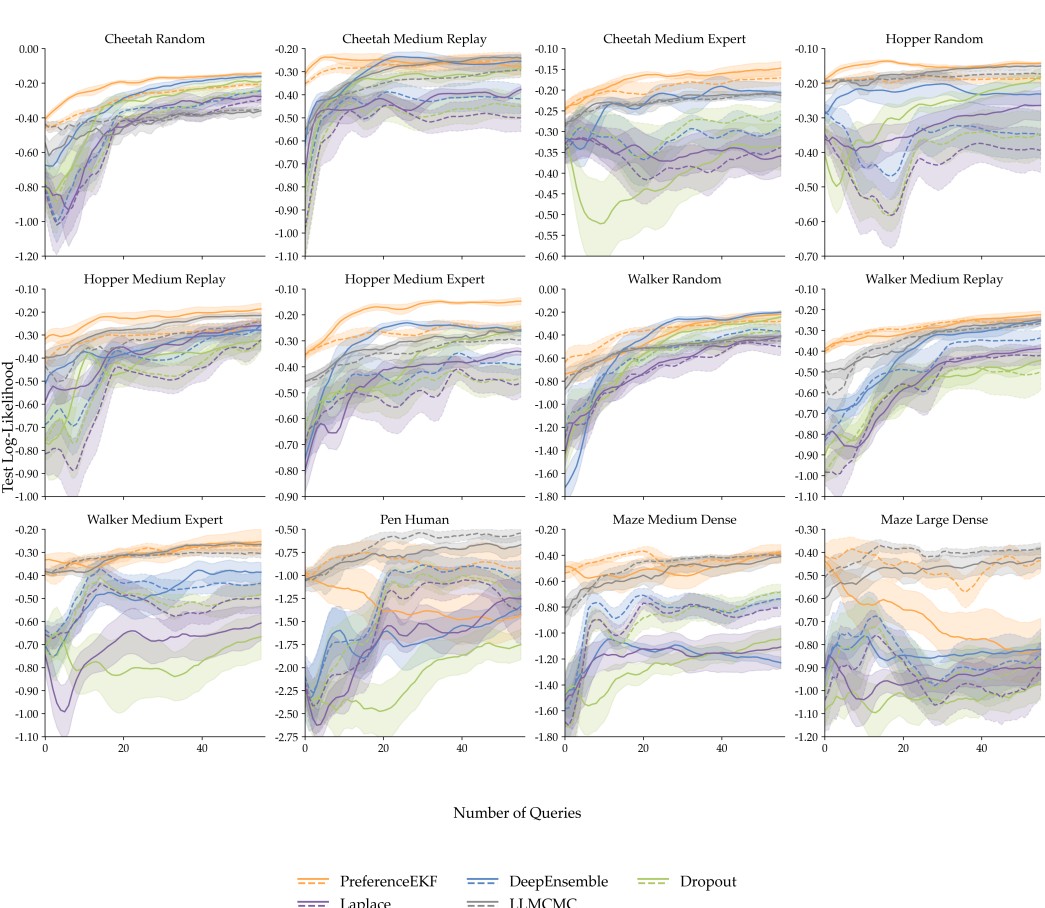

Figure A.3: Comparison of the random (dashed line) and active (solid line) variants of the algorithms using the disagreement acquisition function, across 12 D4RL tasks for preference-based reward modeling (mean±s.e. over 5 seeds). In most tasks, active PreferenceEKF either performs on par with or outperforms other algorithms in terms of sample-efficiency and final log-likelihood. Pen Human are Maze Large Dense are particular outlier cases where active PreferenceEKF severely underperforms, which explains why the aggregate results in Fig. A.2 look unfavorably for active PreferenceEKF relative to its random variant.

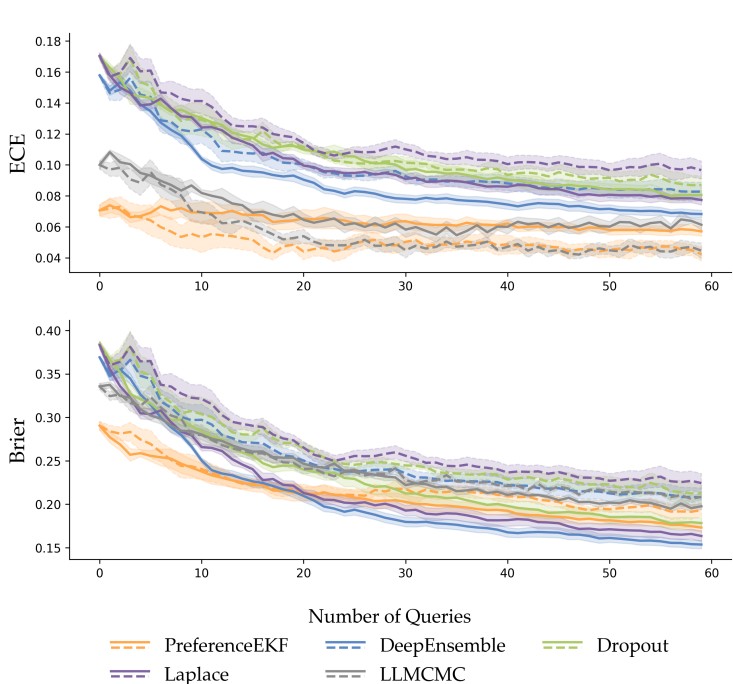

Figure A.4: Comparison of the random (dashed line) and active (solid line) variants of the algorithms in model calibration, as evaluated by expected calibration error and Brier score on a test dataset (lower is better for both metrics).

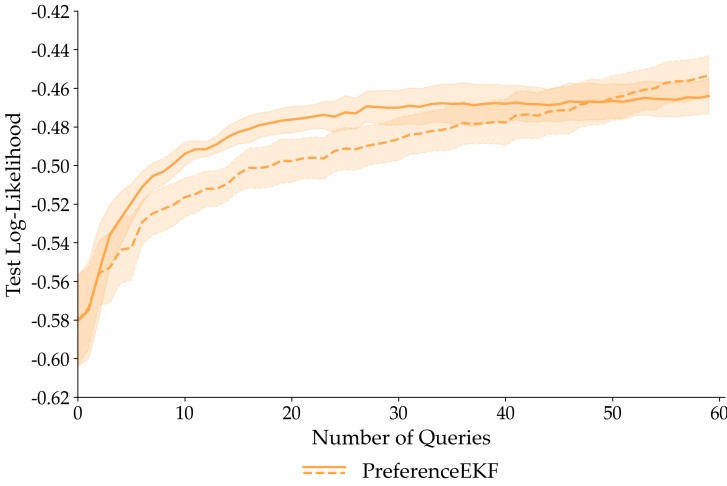

Figure A.5: Comparison of the random (dashed line) and active (solid line) variant of PreferenceEKF for preference-based reward modeling using the InfoGain acquisition function, aggregated over 3 pixel-based VD4RL tasks (mean±s.e. over 5 seeds).

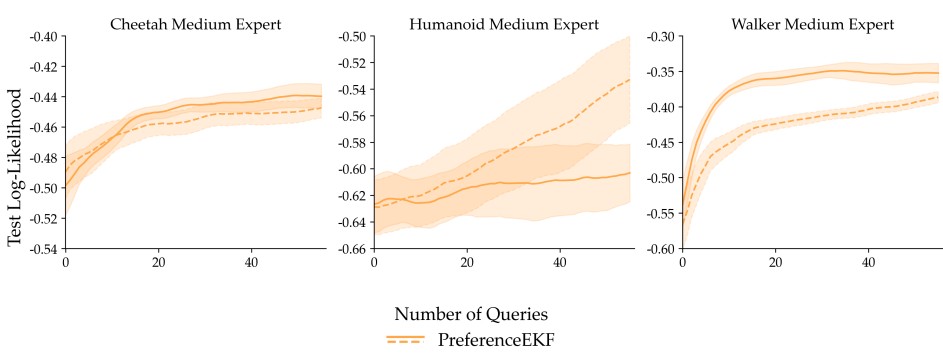

Figure A.6: Comparison of the random (dashed line) and active (solid line) variant of PreferenceEKF for preference-based reward modeling using the InfoGain acquisition function, aggregated over 3 pixel-based VD4RL tasks (mean±s.e. over 5 seeds).

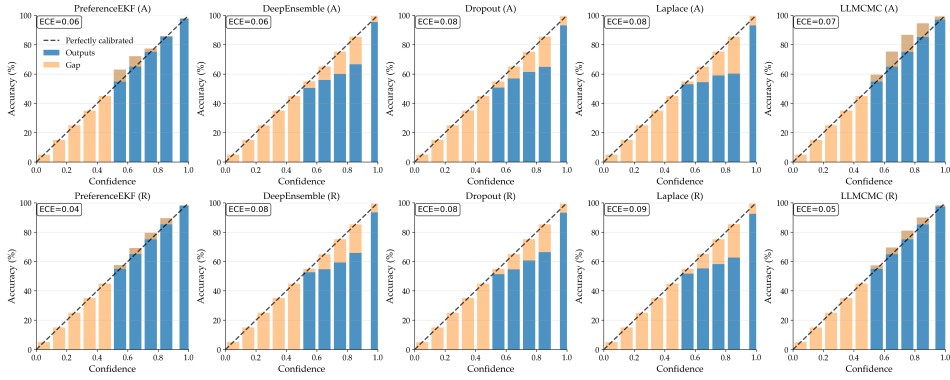

Figure A.7: Reliability diagram over all five methods and their random and active variants.

Following the experiment methodology of Shin et al. (2022) for our offline RL experiments, we add two reference performance scores to every task as shown in Fig. A.8: we refer to "GT" as the score from an offline RL policy trained on $\mathcal{D}^{traj}$ labeled with ground-truth environment reward information, and "Zero" as score from a policy trained on $\mathcal{D}^{traj}$ with reward information zeroed out. This serves to test whether an offline RL algorithm is able to effectively leverage reward information for a given trajectory dataset. For most tasks, GT and Zero serve as upper and lower performance bounds for learned policies.

All offline RL experiments were done by running implicit Q-learning (IQL) (Kostrikov et al., 2021) on trajectory transition datasets labeled with different types of rewards, e.g., ground truth environment reward, zeroed out reward, or preference-learned reward. An IQL agent consists of four neural networks: main and target Q-network, Gaussian policy network, and state-value network. All four networks have two hidden layers of 256 units each and are trained using the same optimizer configuration with cosine decay learning rate schedule. All training runs are done using 1M update steps with 5 rollouts every 50K steps for evaluation. We apply normalization to both reward and observation features, and further apply clipping for reward values exceeding 10. All hyperparameters are detailed in Table 2.

Table 2: Shared hyperparameters for IQL across all tasks. Here "Iterations" refers to the number of minibatch updates.

| Name | Value |
|------|-------|
| Optimizer | Adam |
| Learning rate | 0.0003 |
| Betas | (0.9, 0.999) |
| Iterations | 1M |
| Batch size | 256 |
| Discount factor $\gamma$ | 0.99 |
| Target net update step size | 0.005 |
| Expectile $\tau$ | 0.7 |
| Advantage temperature $\beta$ | 3.0 |
| Exponential advantage clip | 100 |

## A.4 SCALING EXPERIMENTS.

JAX offers efficient vectorization of arbitrary functions using `jax.vmap`. While we use this to parallelize ensemble model training and prediction in most experiments in Section 5, we do not use this for the scalability experiments in Section 5.3. Parallelized training and prediction of up to $M = 150$ models with up to 2M parameters (in the case of the three layer neural networks with 1024 units each) can quickly lead to out-of-memory errors. We instead use python's native for loop to perform ensemble model training and prediction sequentially. All scalability experiments were done on CPU instead of GPU to avoid out-of-memory errors.

## A.5 LLM USAGE

We used LLMs primarily for writing Python visualization scripts, figures/tables typesetting in Latex, finding related work on subspace construction methods, and debugging JAX compilation / model loading errors. We did not use LLMs for paper writing, research ideation, or implementing the core algorithm parts.

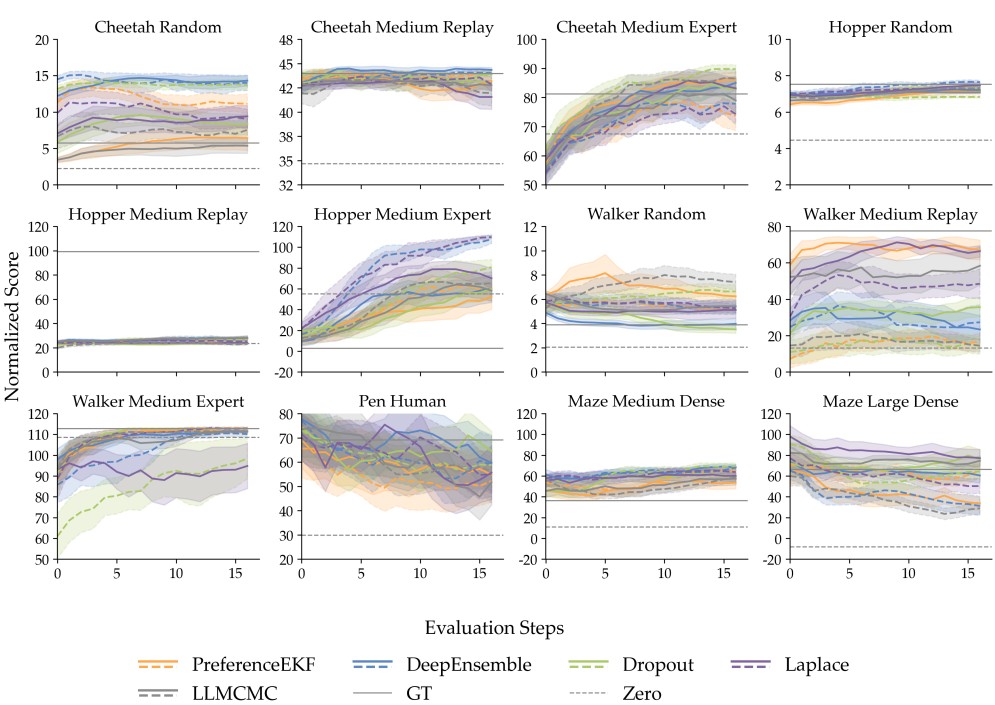

Figure A.8: Comparison of the RM learned using random (dashed line) and active (solid line) variants of the algorithms across 12 D4RL tasks in the offline RL setting (mean±s.e. over 5 seeds). Black solid line indicates the performance of a policy trained on ground truth reward (GT), and black dotted line for a policy trained without reward information (Zero). In most tasks, active PreferenceEKF performs on par with other algorithms in terms of rollout score.

