# OpenReview forum: "Subspace Inference Enables Active Preference based Learning of Neural Network Reward Models"
_ICLR.cc/2026/Conference — Submitted to ICLR 2026_

### Official Review · Reviewer_5nbK · 2025-10-19

**Soundness:** 2
**Presentation:** 1
**Contribution:** 1
**Rating:** 2
**Confidence:** 3

**Summary:**

This paper addresses sample efficiency in reward modeling for RLHF. The authors propose a Bayesian approach to select queries that maximize mutual information between labels and reward model parameters. To ensure practical applicability, the method integrates Bayesian learning with subspace inference, limiting computation to a subset of parameters. Experimental validation on 12 D4RL tasks shows the proposed method achieves competitive performance compared to dropout and ensemble baselines while requiring substantially fewer active labels.

**Strengths:**

1. The paper presents a novel method by applying Bayesian methods to select queries with maximum information gain for active preference labeling.

**Weaknesses:**

1. The paper lacks clarity on the underlying mechanism by which the Bayesian-based method achieves superior sample efficiency. The theoretical or empirical justification for why this approach outperforms alternatives is insufficient.
2. The choice of test-set log-likelihood as an evaluation metric for reward model quality is inadequately justified. The claim of improved sample efficiency rests on showing that the proposed method reaches equivalent test-set log-probability with fewer annotated samples than random sampling. This conclusion requires more rigorous empirical analysis and validation.
3. The baseline comparison is limited, incorporating only dropout and ensemble methods. The evaluation would benefit from including additional state-of-the-art active learning baselines.

**Questions:**

1. In Figure 1, why do methods using random sampling exhibit significant performance variation according to your evaluation metric? Additionally, why does the normalized score for random sampling sometimes exceed that of active labeling with your proposed method?

---

> ### Author Response · Authors · 2025-11-27
>
> We thank the reviewer for the critiques.
>
> > The paper lacks clarity on the underlying mechanism by which the Bayesian-based method achieves superior sample efficiency. The theoretical or empirical justification for why this approach outperforms alternatives is insufficient.
>
> Our method is based on information-theoretic active learning,  which achieves superior sample efficiency by prioritizing training on samples that maximally reduces uncertainty over the maximum likelihood estimate of model weights. In general, this is a highly theoretical question that has roots in fields as diverse as Bayesian experiment design and information theory. We refer the reviewer to classic texts such as [1,2,3] for further details.
>
>
>
> > The baseline comparison is limited, incorporating only dropout and ensemble methods. The evaluation would benefit from including additional state-of-the-art active learning baselines.
>
> We updated the main results with two additional approximate Bayesian methods, last-layer MCMC and Laplace approximation. Please refer to Figures 1 & b + Sections 5.1 & 5.2 for main experiment results on loglikelihood and RL. We further provide description and rationale of the baselines in Appendix A.2.1, for a total of 4 baseline methods to compare to PreferenceEKF.
>
>  To further show that our approach can scale to more complex tasks and architectures, we provide new results on pixel-based tasks. We added results for Visual D4RL (V-D4RL, pixel-based counterpart to D4RL), using a new ResNet18-based reward model architecture. We employed various design choices as detailed in Appendix section A.2.3 to scale PreferenceEKF to pixel-based tasks and significantly larger model architectures.
>
> In particular, our pixel-based tasks use a a new reward model consisting of 13M parameters for a ResNet18 backbone + 200k parameters for a reward predictor head. We reduce this model to subspace of dimension 500 for EKF inference. We hope this address the reviewer’s concern regarding lack of comparison to state of the art approaches.
>
>
>
> > The choice of test-set log-likelihood as an evaluation metric for reward model quality is inadequately justified. The claim of improved sample efficiency rests on showing that the proposed method reaches equivalent test-set log-probability with fewer annotated samples than random sampling. This conclusion requires more rigorous empirical analysis and validation.
>
>  We would like to bring to the reviewer's attention that test-set log likelihood is a standard and widely used metric [4,5,6] for uncertainty quantification and active learning. To provide further rigor to our claims, we provide hypothesis testing with paired tests, 95% confidence, and effect sizes in Appendix A.1. For model calibration results, we refer to the updated Figure 3a and section 5.4. We further add reliability diagram in Appendix A.2.4.
>
>
> [1]  T. Rainforth, A. Foster, D. R. Ivanova, and F. B. Smith, “Modern Bayesian Experimental Design,” _Statistical Science_, vol. 39, no. 1, pp. 100–114, Feb. 2024, doi: [10.1214/23-STS915](https://doi.org/10.1214/23-STS915).
>
> [2]  J. C. Principe, _Information Theoretic Learning: Renyi’s Entropy and Kernel Perspectives_. in Information Science and Statistics. New York, NY: Springer, 2010. doi: [10.1007/978-1-4419-1570-2](https://doi.org/10.1007/978-1-4419-1570-2).
>
> [3]  R. Garnett, _Bayesian Optimization_. Cambridge University Press, 2023.
>
> [4] Y. Ovadia _et al._, “Can You Trust Your Model’s Uncertainty? Evaluating Predictive Uncertainty Under Dataset Shift,” in _Proceedings of the 33rd International Conference on Neural Information Processing Systems_, Red Hook, NY, USA: Curran Associates Inc., Dec. 2019, pp. 14003–14014.
>
> [5]  J. Harrison, J. Willes, and J. Snoek, “Variational Bayesian Last Layers,” presented at the The Twelfth International Conference on Learning Representations, Oct. 2023. Accessed: Feb. 06, 2025. [Online]. Available: [https://openreview.net/forum?id=Sx7BIiPzys](https://openreview.net/forum?id=Sx7BIiPzys)
>
> [6] E. Bıyık, M. Palan, N. C. Landolfi, D. P. Losey, and D. Sadigh, “Asking Easy Questions: A User-Friendly Approach to Active Reward Learning,” in _Proceedings of the Conference on Robot Learning_, PMLR, May 2020, pp. 1177–1190. Accessed: May 13, 2025. [Online]. Available: [https://proceedings.mlr.press/v100/b-iy-ik20a.html](https://proceedings.mlr.press/v100/b-iy-ik20a.html)

---

### Official Review · Reviewer_rzxR · 2025-10-29

**Soundness:** 2
**Presentation:** 3
**Contribution:** 1
**Rating:** 2
**Confidence:** 3

**Summary:**

Reinforcement learning from human feedback (RLHF) relies on preference labels provided by humans, which is time-consuming. Following the line of active learning, this paper proposes leveraging Bayesian filtering in neural network subspaces to efficiently maintain a model posterior for active reward modeling. Compared with ensemble and dropout methods, PreferenceEKF demonstrates superior sample efficiency on the D4RL benchmark.

**Strengths:**

- The paper is well-structured and easy to follow.

- Adopting extended Kalman filters for training neural networks is intuitively sound.

- The experiments demonstrate performance gains compared with ensemble and dropout methods.

**Weaknesses:**

- The proposed method is an adaptation of existing methods for network training. However, some assumptions remain unverified in the experiments, making it unclear how well the method can scale. For instance, the authors cite a paper to support the claim that "neural networks are overparameterized and that solutions actually live in a much smaller subspace." Yet, the phrase "much smaller subspace" is vague, and the values of $|z|$ and $|\theta|$ used in the experiments do not differ significantly.

- One motivation mentioned is that "ensemble methods and dropout are computationally expensive at scale." However, the experiments are tested on a toy model with only two layers, and there are no results demonstrating the proposed method’s performance at scale. This means the experiments fail to support the broader claims introduced in the paper.

- The paper conflates sample efficiency and time efficiency. Conducting experiments with a small network and comparing against only two very basic baselines does not clearly validate either type of efficiency. Additionally, several RLHF or PBRL methods—though not within the active learning paradigm—could have been included as baselines to enable a more comprehensive comparison of sample efficiency [1, 2].

[1]Park, Jongjin, et al. "Surf: Semi-supervised reward learning with data augmentation for feedback-efficient preference-based reinforcement learning." arXiv preprint arXiv:2203.10050 (2022).

[2] Liu, Runze, et al. "Meta-reward-net: Implicitly differentiable reward learning for preference-based reinforcement learning." Advances in Neural Information Processing Systems 35 (2022): 22270-22284.

**Questions:**

- Following your claim that "neural networks are overparameterized and that solutions actually live in a much smaller subspace," I am curious: if we use ensemble methods with N models—each having $|\theta|/N$ parameters, would such an approach achieve good performance and be efficient?

- Please refer to the "Weaknesses" section.

---

> ### Author Response · Authors · 2025-11-27
>
> We thank the reviewer for the positive comments and helpful critiques!
>
> >- For instance, the authors cite a paper to support the claim that "neural networks are overparameterized and that solutions actually live in a much smaller subspace." Yet, the phrase "much smaller subspace" is vague, and the values of |z| and $\theta$ used in the experiments do not differ significantly.
> >- One motivation mentioned is that "ensemble methods and dropout are computationally expensive at scale." However, the experiments are tested on a toy model with only two layers, and there are no results demonstrating the proposed method’s performance at scale. This means the experiments fail to support the broader claims introduced in the paper.
>
> We provide new results on pixel-based tasks to show that our method can scale to more complex tasks and architectures. We added results for Visual D4RL (V-D4RL, pixel-based counterpart to D4RL), using a new ResNet18-based reward model architecture. We employed various design choices as detailed in Appendix section A.2.3 to scale PreferenceEKF to pixel-based tasks and significantly larger model architectures.
>
> In particular, our pixel-based tasks use a a new reward model consisting of 13M parameters for a ResNet18 backbone + 200k parameters for a reward predictor head. We reduce this model to subspace of dimension 500 for EKF inference. We hope this address the reviewer’s concern about the small difference between the full model parameter count and the subspace dimension, as well as scalability to larger models.
>
> > The paper conflates sample efficiency and time efficiency. Conducting experiments with a small network and comparing against only two very basic baselines does not clearly validate either type of efficiency. Additionally, several RLHF or PBRL methods—though not within the active learning paradigm—could have been included as baselines to enable a more comprehensive comparison of sample efficiency [1, 2].
>
> We updated the main results with two additional approximate Bayesian methods, last-layer MCMC and Laplace approximation. Please refer to Figures 1 & b + Sections 5.1 & 5.2 for main experiment results on loglikelihood and RL. We further provide description and rationale of the baselines in Appendix A.2.1, for a total of 4 baseline methods to compare to PreferenceEKF.
>
> We additionally provide hypothesis testing with paired tests, 95% confidence, and effect sizes in Appendix A.1. For model calibration results, we refer to the updated Figure 3a and section 5.4 for results using the new baselines. We further add reliability diagram in Appendix A.2.4.
>
> We see the novelty of our approach as primarily based on being able to learn a high dimensional distribution over large neural network models, which enables sampling of arbitrary number of model weights from the learned posterior distribution, enabling the usage of state of the art acquisition functions such as InfoGain that have only been previously applied to smaller scale models such as linear models. While the references that the authors provided are indeed strong active learning methods, they tackle the problem from fundamentally different approaches (data augmentation and meta learning). We see those approaches as complementary to our method as opposed to competitive. We do see other efficient Bayesian deep learning methods as competitive methods, and hence included them as part of the main results.

---

### Official Review · Reviewer_yWzc · 2025-10-30

**Soundness:** 3
**Presentation:** 3
**Contribution:** 2
**Rating:** 4
**Confidence:** 3

**Summary:**

The paper proposes PreferenceEKF, which maintains a Gaussian posterior over reward-model parameters in a learned low-dimensional subspace and updates it online via an extended Kalman filter (EKF) under a Bradley–Terry preference likelihood. Sampling from this subspace posterior enables efficient, posterior-based active learning via mutual information (MI) without training large ensembles or relying on dropout. Experiments on D4RL tasks report improved label efficiency over random querying, competitive or better calibration, scalable inference, and competitive downstream offline RL returns.

**Strengths:**

Principled active learning: Uses MI over pairwise preferences with a maintained posterior, which is solid Bayesian practice for label efficiency.

Scalable posterior sampling: Subspace + EKF yields cheap sampling for acquisition, avoiding K-way ensembles or heavy MCMC; practical for neural reward models.

Clarity of core idea: The subspace mapping and EKF update are presented clearly; ablations on subspace dimension aid interpretation.

Potential impact: Addresses a central RLHF bottleneck (costly preference labels) by making posterior-based acquisitions feasible.

**Weaknesses:**

Missing Bayesian baselines: Does not compare against competitive and computationally efficient approximate Bayes methods (Laplace redux, last-layer VI, SGLD/SWAG). This weakens novelty and empirical support.

Posterior expressivity: A unimodal Gaussian in a learned subspace with first-order linearization can misrepresent uncertainty under misspecification, multi-modality, or annotator heterogeneity; these regimes are not stress-tested. (More reasoning: The posterior is a single Gaussian in a learned subspace θ=Az+θ\* where A comes from SVD of SGD iterates and the prior in z is isotropic. This is an empirical-Bayes choice whose geometry is optimizer-induced; isotropy in z becomes anisotropy in θ, and coverage depends on whether the iterate subspace spans high-mass posterior directions. There is no evidence-based selection of ∣z∣ (e.g., WAIC/PSIS-LOO) and no hierarchical/shrinkage prior on z. Meanwhile, the iterated EKF with a Bradley–Terry likelihood (n_linearize=5) implements a sequential Laplace/ADF update. It is reasonable near a single mode with mild curvature, but can under-estimate uncertainty under sharp curvature/skew and cannot represent multimodality (e.g., heterogeneous annotators), which can in turn bias MI-based query selection.)

Statistical reporting: Few seeds and limited uncertainty reporting; lacking confidence intervals, effect sizes, and paired tests at matched budgets.

Human data gap: Reliance on synthetic preferences leaves external validity to real annotators unclear.

Compute transparency: Per-query acquisition budget (number of posterior samples M, candidate-pool size/refresh, wall-clock) is under-specified, making it hard to judge real-world efficiency.

**Questions:**

Results for Laplace (full/last-layer), variational last-layer, and SGLD/SWAG on a representative subset?

Exact acquisition budget per query: M, pool size/refresh, and wall-clock on your stated hardware?

Sensitivity to ∣z∣, SVD-iterate selection, and offset θ\*; can you provide evidence-based subspace selection (WAIC/PSIS-LOO)?

Robustness with human or heterogeneous annotators; any hierarchical BT results?

Can you add 95% CIs, paired tests, and effect sizes; plus NLL and reliability diagrams with ECE binning details?

Do conclusions hold under CQL or TD3+BC given the same learned rewards?

---

> ### Author Response · Authors · 2025-11-27
>
> We thank the reviewer for the positive comments and helpful critiques!
>
> >- Results for Laplace (full/last-layer), variational last-layer, and SGLD/SWAG on a representative subset?
> >- Can you add 95% CIs, paired tests, and effect sizes; plus NLL and reliability diagrams with ECE binning details?
>
> We updated the main results with two additional approximate Bayesian methods, last-layer MCMC and Laplace approximation. Please refer to Figures 1 & b + Sections 5.1 & 5.2 for main experiment results on loglikelihood and RL. We further provide description and rationale of the baselines in Appendix A.2.1, for a total of 4 baseline methods to compare to PreferenceEKF.
>
> We provide hypothesis testing with paired tests, 95% confidence, and effect sizes in Appendix A.1
>
> For model calibration results, we refer to the updated Figure 3a and section 5.4 for results using the new baselines. We further add reliability diagram in Appendix A.2.4.
>
> In addition, we provide new results on pixel-based tasks to show that our method can scale to more complex tasks and architectures. We added results for Visual D4RL (V-D4RL, pixel-based counterpart to D4RL), using a new ResNet18-based reward model architecture. We employed various design choices as detailed in Appendix section A.2.3 to scale PreferenceEKF to pixel-based tasks and significantly larger model architectures.
>
> >Exact acquisition budget per query: M, pool size/refresh, and wall-clock on your stated hardware?
>
> Please refer to appendix A.2 for budget query and pool size. We also have experiments quantifying wallclock in Section 5.3 and Figure 2.
>
> > Do conclusions hold under CQL or TD3+BC given the same learned rewards?
>
> We would like to bring to the reviewer’s attention that implicit Q-learning (IQL) is a widely-used state of the art offline RL algorithm. Works have shown [1,2] that CQL, TD3+BC and IQL all achive about the same level of policy rollout performance given the same environment and reward structure. The main result of our paper is to show improved sample-efficiency in active learning settings using PreferenceEKF, so we believe that having just one representative RL method to test out the quality of the learned reward is sufficient.
>
> > Sensitivity to ∣z∣, SVD-iterate selection, and offset θ*; can you provide evidence-based subspace selection (WAIC/PSIS-LOO)?
> >
>
> We provide in Figure 3b and Section 5.5 an ablation to sensitivity of |z| and the susbspace construction technique (random projection vs. SVD). We conclude that larger subspace dimensions (e.g. >500) would be agnostic to the construction technique, while smaller subspace dimensions benefit more from SVD-based construction. Based on this result, for our pixel-based experiments using the much larger ResNet18-based reward model (13M parameters), we use subspace dimension of 500 with random projection-based construction as it is computational cheaper than SVD, which requires storing SGD iterates. While this selection process is not WAIC/PSIS-LOO, it was sufficient for our tasks.
>
>
> > Robustness with human or heterogeneous annotators; any hierarchical BT results?
>
> We once again thank the reviewers for the greate theoretical insight. Since PreferenceEKF is built around the unimodal Gaussian estimation procedure of EKF, we have doubts over the robustness of PreferenceEKF to settings with heterogenous annotators. We believe that some kind of variational inference -based approach using inferred latent per annotator to learn conditional reward models would be a viable approach to this setting, but it is beyond the scope of our work.
>
>
>
> [1] D. Tarasov, V. Kurenkov, A. Nikulin, and S. Kolesnikov, “Revisiting the minimalist approach to offline reinforcement learning,” in _Proceedings of the 37th International Conference on Neural Information Processing Systems_, in NIPS ’23. Red Hook, NY, USA: Curran Associates Inc., Dec. 2023, pp. 11592–11620.
>
> [2] M. Nakamoto, O. Mees, A. Kumar, and S. Levine, “Steering Your Generalists: Improving Robotic Foundation Models via Value Guidance,” presented at the 8th Annual Conference on Robot Learning, Sept. 2024. Accessed: July 24, 2025. [Online]. Available: [https://openreview.net/forum?id=6FGlpzC9Po](https://openreview.net/forum?id=6FGlpzC9Po)

---

### Official Review · Reviewer_8PzR · 2025-11-01

**Soundness:** 3
**Presentation:** 3
**Contribution:** 3
**Rating:** 8
**Confidence:** 2

**Summary:**

This paper is motivated by the poor sample efficiency of reinforcement learning from human feedback (RLHF) and the lack of scalable methods for active learning based mitigations. To address this issue, adopting recent advances in Bayesian subspace filtering, the authors propose the PreferenceEKF (Preference Extended Kalman Filter) method which maintains a posterior distribution over reward model (RM) parameters and enables sampling-based acquisition functions such as InfoGain to perform active learning at scale.

The key contributions are:

- PreferenceEKF, to the best of my knowledge also, the first method to apply Bayesian subspace filtering for active preference-based reward learning.

- Scaling InfoGain for active preference-based reward learning from linear models to neural networks.

- Showing empirically, using the D4RL offline RL benchmark, that PreferenceEKF outperforms DeepEnsemble and Dropout baselines in terms of sample efficiency (~42% fewer samples), scalability (Figure 2) and model calibration (Figure 3b) whilst  maintaining or slightly exceeding in trained policy performance.

- An open source implementation.

**Strengths:**

The key strength of this paper, contributing to both its originality and significance, is the PreferenceEKF method which is the first to apply Bayesian subspace filtering for active preference-based reward learning. By extending EKF in a low-dimensional subspace the $\mathcal{O}(|{\theta}|^2)$ posterior scaling is avoided, enabling the use of InfoGain for active preference-based reward learning to scale from linear models to neural networks.

The clarity of the work is high and it’s clear how and why PreferenceEKF enables sampling from high-dimensional distributions to scale InfoGain to neural network models. The experiments are high-quality (e.g., well-structured research questions, appropriate baselines and task selection, multiple runs, hyperparameters reported, ablation analysis etc) and the resulting discussion is also insightful. The limitations are clearly and (to the best of my understanding) faithfully discussed.

I think the potential significance of this work is high given the limitations of linear reward models and the importance of active preference-based reward learning.

**Weaknesses:**

As noted by the authors, it remains to be determined whether PreferenceEKF will scale to foundation model-scale reward models and this is not considered within the current paper (in fact the models trained are modest at 2x64 - 2x256 hidden units). The evaluation, although thorough, is also limited to the D4RL benchmark. It would be interesting if more could be said about the Pen Human outlier case where active PreferenceEKF "severely underperforms".

The work could be improved further by establishing theoretical bounds or guarantees on Bayesian subspace filtering for active preference-based reward learning. Where this is not possible in the scope of the current work it would nevertheless benefit from signposting the most closely-related or promising theoretical foundations.

Minor:

- I don’t think the ‘RM’ acronym is explicitly defined anywhere (reward model?)

- line 226: the footnote (3) is hanging between the end of the previous sentence and the start of the next.

- In A.3 you mention "in the case of the three layer neural networks with 1024 units each" but this seems to be the only reference to such a case.

**Questions:**

1. Are you able to offer any insights about the Pen Human outlier case where active PreferenceEKF "severely underperforms"?

2. Can you please comment on how significant the jump from linear reward models to modest neural networks is in contrast to foundation-model sized reward models?

---

> ### Author Response · Authors · 2025-11-27
>
> We thank the reviewer for the positive comments and helpful critiques!
>
> To address the critique on benchmark diversity and scalability to bigger reward model architectures, we added results for Visual D4RL (V-D4RL, pixel-based counterpart to D4RL), using a new ResNet18-based reward model architecture. We employed various design choices as detailed in Appendix section A.2.3 to scale PreferenceEKF to pixel-based tasks and significantly larger model architectures.
>
> > Can you please comment on how significant the jump from linear reward models to modest neural networks is in contrast to foundation-model sized reward models?
>
> In previous works such as [1], the number of parameters in a linear model scales with the dimensionality of the task’s observation space (<10). In our state-based experiments, our base neural network reward model has <10k parameters reduced to subspace dimension of 200. For V-D4RL, the ResNet18-based reward model has 13M parameters, of which around 200K (reward head, excluding ResNet backbone) are reduced to subspace dimension of 500 for inference. The current largest reward models are based on LLMs [2], and has paramter count up in the billions.
>
> > Are you able to offer any insights about the Pen Human outlier case where active PreferenceEKF "severely underperforms"?
>
> In lieu of rigorous experimentations, we hypothesize that the acquisition function in question, disagreement, favors queries that are difficult for the model in the “adversarial” sense, rather than in the “immediately helpful to learn from” case. This can result in an aggregated dataset that becomes progressively more skewed, resulting in model overfitting or miscalibration. Our main results use the InfoGain acquisition function, and we do not observe any "severely underperforming" behaviors across all tasks in appendix Figure A.1
>
> [1] E. Bıyık, M. Palan, N. C. Landolfi, D. P. Losey, and D. Sadigh, “Asking Easy Questions: A User-Friendly Approach to Active Reward Learning,” in _Proceedings of the Conference on Robot Learning_, PMLR, May 2020, pp. 1177–1190.
>
> [2] D. Mahan _et al._, “Generative Reward Models,” Oct. 02, 2024, _arXiv_: arXiv:2410.12832.

---

### Author Response · Authors · 2025-12-03
**Author final remarks**

We thank the AC and all reviewers for their time and thoughtful feedback. The comments greatly helped us improve the paper. Here we provide a summary of the strengths and weaknesses mentioned by each reviewer, and how we addressed them.

Reviewer `8PzR`
- Praised our paper for its novelty, clarity in writing, well formulated and executed experiments, and good discussions of results and potential pitfalls. They also praised our paper for its significance.
- Critiqued our paper for the lack of additional benchmarks, the limited model architecture size, and lack of further discussion for one particular experiment run (Pen Human, which is ran with a baseline acquisition method, disagreement) other than acknowledgement of its weaker performance. They also suggested adding work on theoretical bounds in the active learning setting.
- We addressed the critiques by adding adding a new pixel-based benchmark (V-D4RL), and further tested our methods on a much larger model architecture for the new task (ResNet backbone reward model, with 13M parameters, of which 200K are trainable). We also offered our hypothesis on the Pen Human experiment’s poor performance, and commented on the applicability of our method for LLM-sized models.

Reviewer `yWzc`
- Praised our paper for its principled active learning approach, its scalability, clarity of writing, and significance.
- Critiqued our paper for the lack of additional state-of-the-art (SOTA) Bayesian deep learning baselines, the lack of statistical testing (beyond the 95% confidence intervals we provided).
	- They further critiqued our method’s inability to acommodate heterogenous preference learning settings with multiple annotators (particularly in the real world case, not just preference data from simulated domains like D4RL).
	- Lastly, they asked for further implementation details, and for evidence-based subspace selection using methods such as WAIC / PSIS-LOO.
- We addressed the critiques by adding two additional SOTA baselines, with corresponding active learning, offline RL, and model calibration experiments, as well as discussion for why we chose those baselines. We also added statistical significance testing for our results, using 1-sided paired t-test with 95% confidence intervals and effect sizes. We observed strong statistical significance of our main results, as detailed in Appendix A.1.
	- We further provided discussions on implementation details such as query budget, wallclock details, other offline RL algorithms, and sensitivity to subspace dimensionality.
	- We provided our take on heterogenous preference learning. While we remind that the weakness was already acknowledge in our paper and may be beyond the current scope of our work, we also provided our hypothesis on how we can tackle this problem in a future iteration of our work.

Reviewer `rzxR`
- Praised our paper for its clarity in writing, strong empirical results, and sound idea.
- Critiqued our paper for the experiments’ limited model architecture size, the relatively smaller difference between the full parameter space dimensionality versus the subspace dimensionality, and the lack of additional SOTA baselines in both Bayesian deep learning and preference-based RL.
- We addressed the critiques by adding the V-D4RL benchmark, and further showed our methods’ applicability on the much larger ResNet-based reward model we designed. We also added two strong Bayesian deep learning baselines, and showed our statistical significance of our method outperforming all four baselines.
	- We provided discussion on why it would be more fair to compare our method to the Bayesian deep learning baselines as opposed to the preference-based RL baselines.

Reviewer `5nbK`
- Praised our paper for its novelty.
- Critiqued our paper for lack of discussion on why Bayesian methods are sound for active learning, for our choice of the test-set loglikelihood metric, and lack of baselines.
- We addressed the critiques by adding two state of the art Bayesian deep learning baselines, and showing strong statistical significance in our main results.
	- We provided discussion on the intuition of the Bayesian approach to information-theoretic active learning. We emphasize that this question is at the core of the field of active learning, and provided additional pointers to classic works that answer this question. The core scope of our paper is not to answer this question or establish theory on why it works, but to provide an efficient and effective method that builds upon previous works in the field.
	- We also point out that test-set log-likelihood is a widely used metric in active learning literatures, and provided a handful of works that justify its usage.

We believe we have addressed all of the reviewers’ concerns, and have updated our paper with all additional experiments, benchmarks, baselines, statistical testing, implementation details, and discussion. Thank you all again for your thoughtful engagement with our work.

---

### Meta-Review · Area_Chair_aE1N · 2026-01-08

**Summary:**

The paper proposes PreferenceEKF, a Bayesian approach to active preference-based reward learning that maintains a posterior distribution over neural network reward model parameters under a Bradley–Terry preference likelihood using an extended Kalman filter (EKF) in a learned low-dimensional subspace. By then sampling from this distribution, the method then enables efficient, posterior-based active learning via mutual information (MI), without the need to train large ensembles or rely on dropout. The paper evaluates PreferenceEKF on preference-learning and offline reinforcement learning (RL) tasks drawn from the D4RL benchmark, demonstrating improvements in label efficiency, scalable inference, and competitive calibration and downstream performance.

The paper was evaluated by four reviewers, who generally agree that the core idea is principled, that the paper is clear and well-written, and that the work is potentially impactful for RLHF settings where acquiring preference labels is expensive. Some reviewers appreciate the novelty of applying Bayesian subspace filtering to preference-based reward learning. At the same time, some of the reviewers raised concerns regarding the scope of the experimental evaluation. In particular, reviews noted the absence of relevant baselines (including Bayesian, RLHF, and other preference-based reward learning methods) in the paper as originally submitted, that the evaluation is limited to small-scale models, that the paper lacks an adequate evaluation of computational cost, and the use of a limited number of seeds and the lack of uncertainty reporting, which collectively fail to support some of the paper's claims. Additional concerns include the reliance on a unimodal Gaussian posterior, which may inadequately capture the multimodality/heterogeneity of human preferences. These issues led to some divergence in the reviewers' overall recommendations.

**Reviewer Concerns:**

The authors made an effort to address these concerns in their rebuttal and updates to the paper. The authors added new Bayesian deep learning baselines, expanded the experimental evaluation to include pixel-based Visual D4RL tasks with a significantly larger ResNet-based reward model as a means of addressing concerns about model scale, and provided additional ablations on subspace dimensionality and construction. The authors strengthened the statistical analysis by adding paired hypothesis tests and confidence intervals. Additionally, the rebuttal helps to clarify some implementation details, such as acquisition budgets and wall-clock scaling. The authors acknowledged the limitations inherent in using a unimodal (Gaussian) distribution.

**Reviewer Scores:**

While one reviewer argues that the paper be accepted, they rate their confidence as low. While the AC believes that the other reviewers would raise their scores in light of the authors' rebuttal, it is unlikely that the changes would be significant enough to warrant acceptance.

---

### Decision · Program_Chairs · 2026-01-26

Reject